METHODS AND APPROACHES

# Embedding muscle fibers in hydrogel improves viability and preserves contractile function during prolonged *ex vivo* culture

Leander A. Vonk[1]* , Osman Esen[1]* , Daan Hoomoedt[1] , Rajvi M.N. Balesar[1] , Coen A.C. Ottenheijm[1,4] , and Tyler J. Kirby[1,2,3,5]

***Ex vivo*** **culture of isolated muscle fibers can serve as an important model for *in vitro* research on mature skeletal muscle fibers. Nevertheless, this model has limitations for long-term studies due to structural loss and dedifferentiation following prolonged culture periods. This study aimed to investigate how *ex vivo* culture affects muscle fiber contraction and to improve the culture system to preserve muscle fiber morphology and sarcomere function. Additionally, we sought to determine which culture-induced changes can negatively affect muscle fiber contraction. We cultured isolated flexor digitorum brevis (FDB) muscle fibers in several conditions for up to 7 days and investigated viability, morphology, and the unloaded sarcomere shortening in intact fibers, along with force generation in permeabilized muscle fibers. In addition, we examined changes to the microtubule network. We found a time-dependent decrease in contractility and viability in muscle fibers cultured for 7 days on a laminin-coated culture dish (2D). Conversely, we found that culturing FDB muscle fibers in a low-serum, fibrin/Geltrex hydrogel (3D) reduces markers of muscle fiber dedifferentiation (i.e., sprouting), improves viability, and retains contractility over time. We discovered that the loss of contractility of cultured muscle fibers was not the direct result of reduced sarcomere function but may be related to changes in the microtubule network. Collectively, our findings highlight the importance of providing muscle fibers with a 3D environment during *ex vivo* culture, particularly when testing pharmacological or genetic interventions to study viability or contractile function.**

## Introduction

In skeletal muscle research, several *in vitro* models are used to study disease mechanisms and fundamental processes, such as differentiation, myofiber growth and repair, and metabolic signaling (Tarum et al., 2023; Khodabukus et al., 2018; Trendelenburg et al., 2009; Dietze et al., 2002). *In vitro* skeletal muscle cell cultures are powerful models because of their simplicity, renewability, high controllability, and limiting the use of laboratory animals (Khodabukus et al., 2018; Guo et al., 2014; Verma et al., 2020). Common models include the culture of primary cells (Roman et al., 2018), immortalized cell lines (Denes et al., 2019), or induced pluripotent stem cell–derived myoblasts (Bou Akar et al., 2024) and their subsequent differentiation into myotubes. However, these models often fail to replicate the complex architecture of muscle fibers (Khodabukus et al., 2018; Smith and Meyer, 2020), as differentiated myotubes often lack defining features such as a large cross-sectional area (CSA) (Khodabukus et al., 2018; Smith and Meyer, 2020), organized cytoskeleton network (Khodabukus et al., 2018; Smith and

Meyer, 2020; Engler et al., 2004), proper alignment of sarcomeres (Engler et al., 2004) in series and parallel, mature sarcomere proteins (Ravenscroft et al., 2007), fiber type variations (Khodabukus et al., 2018), and junctional regions (Khodabukus et al., 2018; Smith and Meyer, 2020). In particular, *in vitro* studies are limited if the primary outcome is assessing differences in contractile function, thus limiting the ability to translate findings into an *in vivo* context.

*Ex vivo* muscle fiber culture is a viable alternative to traditional methods (Stange et al., 2020; Duddy et al., 2011). In this approach, muscle fibers fully mature *in vivo* before the entire muscle is enzymatically digested to obtain single muscle fibers (Smith and Meyer, 2020; Stange et al., 2020; Duddy et al., 2011; Pegoli et al., 2020). This approach yields hundreds of viable and intact muscle fibers with preserved sarcomere contraction and cellular integrity, making them suitable for contraction-based studies (Garcia-Pelagio et al., 2020; Keire et al., 2013). Despite the potential of *ex vivo* cultures, isolated muscle fibers exhibit

[1]Department of Physiology, Amsterdam University Medical Center, Vrije Universiteit Amsterdam, Amsterdam, The Netherlands; [2]Amsterdam Cardiovascular Sciences, Heart Failure & Arrhythmias, Amsterdam, The Netherlands; [3]Amsterdam Movement Sciences, Tissue Function & Regeneration, Amsterdam, The Netherlands; [4]Department of Cellular and Molecular Medicine, University of Arizona, Tucson, AZ, USA; [5]Division of Cardiovascular Medicine, Department of Internal Medicine, University of Kentucky, Lexington, KY, USA.

*L.A. Vonk and O. Esen contributed equally to this paper.   Correspondence to Tyler J. Kirby: t.kirby@amsterdamumc.nl.

limited viability, with rapid deterioration depending on the culture conditions (Keire et al., 2013; Smith and Meyer, 2020). Studies focused on the serum content of culture media showed that high-serum content (10–20%) has a detrimental effect on muscle fiber morphology and health by promoting the proliferation of satellite cells and dedifferentiation of muscle fibers, while low to serum-free conditions limited satellite cell proliferation and dedifferentiation of muscle fibers (Renzini et al., 2018; Selvin et al., 2015; Rosenblatt et al., 1995; Brown and Schneider, 2002). Using such media compositions, fibers can only be cultured in suspension for up to 3–4 days (Smith and Meyer, 2020; Renzini et al., 2018) unless adhesion is promoted using laminin-coated dishes (Pasut et al., 2013; Guo et al., 2014; Smith and Meyer, 2020; Renzini et al., 2018). Although modulating serum content and promoting adhesion can improve the longevity of muscle fiber culture, these interventions are not enough to preserve muscle fiber morphology for longer than 12 days (Brown and Schneider, 2002).

Long-term studies on *ex vivo* muscle fibers are difficult, and most studies using this model focus on short-term interventions like drug testing (Hoppstadter et al., 2020) or satellite cell activation and muscle fiber regeneration (Renzini et al., 2018; Brun et al., 2018; Feige et al., 2021). The handful of studies that make use of long-term *ex vivo* muscle fiber culture focused on muscle fiber morphological alterations (Renzini et al., 2018; Brown and Schneider, 2002; Ravenscroft et al., 2007), without assessing the effect of long-term culture on contractile function. Furthermore, while some protocols for 3D muscle fiber culture exist (Rausch et al., 2020; Vonk et al., 2023), their effects on contractile performance in both long- and short-term studies remain largely unknown. The development of a long-term culture system that maintains muscle fiber contraction would allow for contraction-based studies at the single-cell level. These include investigations into muscle fiber wasting, the effects of extended pharmacological treatments, and systematic interventions through genetic manipulation using viral approaches.

In this study, we aimed to characterize the contractile function of muscle fibers during prolonged culture and to optimize culturing conditions for muscle fibers to extend their viability while preserving contractile function. We first validated culture medium composition and assessed morphological and functional changes in a 2D culture setup over a longer culture time. We then tested the effect of embedding cells in a fibrin-based hydrogel to mimic the native environment of the muscle fiber. By evaluating fiber morphology, contractility, and sarcomere structure, we investigated the impact of culture duration on *ex vivo* muscle fibers, comparing 2D and 3D conditions. Finally, we aimed to identify factors driving muscle fiber deterioration in culture, which could be modulated to preserve muscle fiber function during prolonged *ex vivo* culture. Our main findings show that muscle fiber function and viability deteriorate over time in culture and that culturing muscle fibers in fibrin-based hydrogel improves viability and contractile function. We further reveal that culture induces changes to the microtubule (MT) network and that the MT network plays an important role in muscle fiber contractile dynamics.

## Materials and methods
### Muscle fiber isolation
Postmortem tissue was obtained from animals sacrificed for other approved research projects and/or the breeding surplus of the VU University in accordance with the European Council Directive (2010/63/EU) by permission of the Animal Research Law of the Netherlands. Isolated mature single muscle fibers were obtained following the protocol described by Vonk et al. (2023). Briefly, adult wild-type C57BL/6 mice from a broad range of genetic backgrounds were obtained from the Amsterdam Animal Research Center. Whole flexor digitorum brevis (FDB) muscles were dissected from the mouse by cutting the tendons. Muscles were transferred to pre-warmed (37°C) and pH-equilibrated (5% $CO_2$) dissection medium, consisting of MEM with high glucose and pyruvate (Gibco; Thermo Fisher Scientific), 10% heat-inactivated FBS (Thermo Fisher Scientific), and 1% penicillin-streptomycin (Sigma-Aldrich). FDB muscles were then transferred to 5 ml of sterilized 0.2% collagenase type II (Worthington Biochemical) dissolved in a dissection medium and digested in a tissue culture incubator at 37°C and 5% $CO_2$ for 70–80 min. After digestion, the whole muscles were carefully transferred to 3 ml of dissection medium and incubated for 30 min in the same conditions. To release the individual muscle fibers, the digested muscle was triturated using two p1000 pipette tips with bore widths of different sizes. Starting with the tip with the widest opening, the muscles were triturated until all fibers were released and in suspension. The tendons were then carefully removed with a regular pipette tip. Myoblasts and remaining tissue were removed through two rounds of gravity sedimentation.

### Muscle fiber culture
Isolated mouse FDB muscle fibers were cultured in a medium consisting of MEM with high glucose and pyruvate (Gibco; Thermo Fisher Scientific) supplemented with 0.4% Serum Replacement 2 (50X) (Sigma-Aldrich), 1% horse serum (Sigma-Aldrich), and 1% penicillin/streptomycin (Sigma-Aldrich). For 2D fiber cultures, culture plates were coated with mouse laminin (Sigma Merck) diluted in MEM at 1:25 for 2 h before seeding to allow adhesion to the culture dish.

To test the effects of different serum concentrations on muscle fiber culture health, muscle fibers were cultured for 7 days in medium of differing compositions. Tested conditions included serum-free Gibco-MEM, Gibco-MEM supplemented with 0.4% serum replacement 2 and 1% horse serum (low serum), and dissection medium containing 10% FBS.

### Fibrin hydrogel preparation and seeding
For the 3D culture condition, a hydrogel was formulated in which fibers could be embedded immediately after isolation and prior to putting them in culture. Hydrogels consisted of 2.5, 3.75, or 5 mg/ml fibrin, to which 10% of either 4 mg/ml Matrigel GFR Membrane Matrix (Matrigel; Corning) or 4 mg/ml Geltrex LDEV-Free Reduced Growth Factor Basement Membrane Matrix (Geltrex; Thermo Fisher Scientific) diluted in Iscove's Modified Dulbecco's Medium + L-glutamine (Thermo Fisher Scientific) was added.

Fibrinogen isolated from bovine plasma (Sigma-Aldrich) was dissolved in PBS (Gibco) at a stock concentration of 20 mg/ml. Fibrinogen solution was then filter-sterilized using a 0.2-µm filter (Whatman), and sterile fibrinogen stocks were stored at −80°C. Thrombin isolated from bovine plasma (Sigma-Aldrich) was dissolved in PBS at a concentration of 125 U/ml and stored at −20°C.

To cross-link fibrin gels, solutions of fibrinogen and thrombin were prepared separately. "Gel solution" consisted of culture medium supplemented with 20% of 4 mg/ml Matrigel or Geltrex and 5–10 mg/ml fibrinogen, depending on the final fibrin concentration. "Cross-linking solution" consisted of culture medium supplemented with 60 µM aprotinin (a fibrinolysis inhibitor to prevent hydrogel degradation) (Thermo Fisher Scientific) and 0.5–1 U/ml thrombin (Thermo Fisher Scientific), depending on fibrinogen concentration. Both solutions were kept on ice before plating to prevent premature polymerization of Matrigel or Geltrex. Muscle fibers were seeded in gels by first allowing fibers to settle to the bottom of a 15-ml tube, after which the supernatant was removed, and cells were resuspended in cross-linking mix. Fibrin gels were cross-linked by adding equal parts of gel solution to cross-linking solution, mixing briefly, and then transferring to a culture plate dish. Gels were then allowed to solidify at 37°C and 5% $CO_2$ in an incubator for up to 2 h. After solidification, gels were completely covered with culture medium. When culturing muscle fibers in gel, 60 µM aprotinin and 200 µM tranexamic acid (Thermo Fisher Scientific) were added to the culture medium to prevent degradation of the fibrin gel.

## Muscle fiber viability and dedifferentiation classification
Culture well plates were sampled at 5 random locations for each condition using a ZOE Fluorescent Cell Imager (Bio-Rad) with a 20× objective. Single-fiber images were then randomized and presented to 11 participants, blinded to the experimental conditions. Participants were asked to classify fiber images according to two predetermined groups: (1) living and (2) dead fibers for the viability classification, or (1) normal and (2) dedifferentiated fibers for the dedifferentiation classification. An example image of representative fibers was shown during classification. Classification of each fiber was then recorded as the majority vote out of all 11 participants.

## Electrical stimulation and contractile measurements
Muscle fibers were stimulated using a 6-well C-Dish insert designed for Corning 24-well plates (IonOptix) coupled to a Myo-Pacer cell stimulator (IonOptix). Electrical stimulations were performed using bipolar pulses at 10 V, a frequency of 1 Hz, and a pulse duration of 4 ms. Muscle fiber contractions were then measured by stimulating fibers inside a CytoCypher MultiCell High-Throughput System (IonOptix). Using the CytoSolver software, sarcomere lengths were measured during contractions at a sampling rate of 250 Hz. From these measurements, a contraction transient is generated, recording parameters such as sarcomere length, contractile velocity, and contraction duration. Data from each fiber are based on the average of 9–10 subsequent contractions. Automatic data analysis was performed using

CytoSolver Transient Analysis Tool software. CytoSolver software automatically excludes single transients that have insufficient $R^2 < 0.95$ fit values. For measurements where transients were rejected by the software, a cutoff of at least four transients was maintained, and data points whose average was based on three or less transients were removed from the dataset. Muscle fiber contractile parameters recorded by the MultiCell system were the following: Sarcomere length at rest, contractile velocity, contraction duration, percentage of sarcomere shortening, relaxation velocity, relaxation duration, and sarcomere length at peak contraction.

## Muscle fiber adhesion contractile measurements
The effects of adhesion during culture were tested by either culturing muscle fibers in 2D as described above or by free-floating culture using uncoated glass-bottom culture plates. A third condition was also included in which muscle fibers were cultured in 2D for 6 days and dissociated from the culture plate by treatment with 0.25% trypsin in EDTA (Gibco; Thermo Fisher Scientific) for 2 min before replating on laminin-coated dishes. All measurements were performed on day 7 after isolation.

## MT network manipulation
To alter the MT network, cells were treated with 20 µM Nocodazole (Cayman Chemical) for 2 h to depolymerize MTs or 100 nM Taxol (Cayman Chemical) overnight to stabilize MTs prior to measurements. The interference with the MT network was confirmed through visual inspection via immunofluorescent labeling for α-tubulin.

## Immunofluorescent staining and image acquisition
Muscle fibers were fixed by replacing half of the culture medium with a solution of 4% paraformaldehyde (PFA, Thermo Fisher Scientific) for 5 min and then a full replacement with 4% PFA for 10 min. PFA was washed out three times for 5 min with PBS and stored for up to 7 days at 4°C. Samples were then blocked and permeabilized using a blocking buffer solution containing 3% BSA (Sigma-Aldrich), 0.2% Triton-X100 (Sigma-Aldrich), and 0.05% Tween (Sigma-Aldrich) dissolved in PBS for 1 h at 4°C on a shaker. Samples were then incubated in a blocking solution with added primary antibodies (diluted according to the manufacturer's instructions, Table 1) overnight at 4°C on a shaker. Samples were then washed with IF buffer containing 0.3% BSA, 0.2% Triton-X100, and 0.05% Tween dissolved in PBS and incubated with a 1:250 dilution of selected AlexaFluor secondary antibodies, 1:1,000 DAPI (Invitrogen) and 1:1,000 AlexaFluor Plus 647 Phalloidin (Invitrogen). The excess staining solution was washed out using PBS, and samples were stored for up to 5 days in PBS at 4°C. Using a confocal microscope (Nikon Ti2 Z-Drive), camera (A21A726014; Prime BSI express), CrestOptics X-Light spinning disc with pinhole 50 µm (Crest X-Light V3), and laserbox (NI DAQ AOTF Multilaser) with line wavelengths of 405, 470, and 555 nm, the muscle fibers were imaged at a magnification of 60× (PLAN APO 60× IOL OFN25 DIC) on a confocal microscope. 1 µm z-stacks of fibers were collected and stacked together using ImageJ to generate maximum-intensity projection images. Mean fluorescence intensity values were

**Table 1.  Antibodies used for immunofluorescent labeling**

| AB ref name | Antigen | Antibody isotype | Dilution | Company |
|---|---|---|---|---|
| A7811 | α-Actinin | MIgG1 | 1:1,000 | Sigma-Aldrich |
| A4.1025 | Myosin | MIgG2a | 1 µg/ml | Developmental Systems Hybridoma Bank |
| 12G10 anti–α-tubulin | α-Tubulin | MIgG1 | 1 µg/ml | Developmental Systems Hybridoma Bank |
| IIH6 C4 | α-Dystroglycan | MIgM | 1 µg/ml | Developmental Systems Hybridoma Bank |
| MANDRA1(7A10) | Dystrophin | MIgG1 | 1 µg/ml | Developmental Systems Hybridoma Bank |

obtained using ImageJ software from maximum-intensity projection images by using phalloidin staining as a mask to outline the myofiber boundaries.

### Tracking of muscle fiber morphology over time

To determine muscle fiber width, length, and sarcomere length changes, muscle fibers were cultured for up to 7 days, and images of the same fiber were collected at days 1 and 7. Measurements were performed using ImageJ software. Sarcomere lengths were measured by measuring 10 sarcomeres and averaging sarcomere length over these 10 sarcomeres. For width and sarcomere length measurements, measurements were taken in three locations and averaged to generate a mean per fiber. To account for the fact that if fiber volume remains constant (i.e., isovolumetric), then sarcomere length and fiber width will change proportionally, we normalized width measurements to the percentage change in sarcomere length of the corresponding fiber. Thus, any observed change in fiber width would indicate a change in fiber volume.

### Permeabilized muscle fiber contraction

We adapted previously described methods for single muscle fiber force measurements (de Winter et al., 2020) to investigate cultured FDB muscle fiber force production after culture. In brief, FDB fibers were cultured in 2D or 3D for 1 and 7 days. Muscle fibers were first treated with 10 µM N-benzyl-p-toluenesulfonamide (BTS; Thermo Fisher Scientific) for 10 min prior to permeabilization to prevent hypercontraction. Then, culture medium was gradually replaced with skinning solution by three 1:1 media changes. Skinning solution consisted of 50% glycerol (Sigma-Aldrich) and 50% relaxing solution containing 100 mM balanced electrolyte solution (Sigma-Aldrich), 6.97 mM ethylene glycol tetraacetic acid (EGTA, Sigma-Aldrich), 6.48 mM MgCl$_2$ (Sigma-Aldrich), 5.89 mM Na$_2$–adenosine triphosphate (ATP; Sigma-Aldrich), 40.76 mM K-propionate (Sigma-Aldrich), 14.5 mM creatine phosphate (Sigma-Aldrich) supplemented proteinase inhibitors 200 µg/ml Leupeptin (Thermo Fisher Scientific), 100 µM L-trans-3-carboxyoxiran-2-carbonyl-L-leucylagmatine (E-64; Sigma-Aldrich), 250 µM phenylmethylsulfonyl fluoride (PMSF; Sigma-Aldrich), 1 mM dithiothreitol (Sigma-Aldrich), and 10 µM BTS. Single muscle fibers were then manually removed from the culture surface or hydrogel using ultrafine forceps, attached to aluminum clips, and measured as described previously (de Winter et al., 2020). Muscle fiber CSA was estimated by measuring muscle fiber width (from a standard top-down view) and height (using a mirror to the side of the fiber) in three separate locations.

### Myosin heavy chain isoform composition

Since the contractile properties of muscle fibers are affected by their myosin heavy chain composition, we investigated changes in myosin heavy chain composition in response to culture using a specialized SDS-PAGE technique as described previously (de Winter et al., 2020). Briefly, muscle fibers were denatured by boiling in SDS sample buffer for 2 min, after which they were loaded onto the gel and run for 24 h at 15°C and a constant voltage of 275. As a control, a combined whole muscle lysate of mouse Soleus and EDL muscles was loaded onto the gel to show the separation of myosin heavy chain isoforms. Finally, gels were stained using SYPRO protein gel stain (Invitrogen), imaged, and analyzed using ImageJ software.

### Statistical analysis

Experimental results from muscle fibers were obtained from at least three separate isolations and three individual mice, unless otherwise stated. For single-fiber experiments, $n$ is denoted as the number of fibers, while $N$ is denoted as the number of mice. Mean values were generated for each mouse, and statistical analysis was performed using the mean value of each mouse. Statistical analysis tests used were either Student's $t$ tests (for datasets with only two conditions), ordinary one-way ANOVA tests (for datasets with more than two conditions) with a Tukey's post hoc test, or two-way ANOVA (for grouped datasets) with a Šidák post hoc test. Statistical analysis of classification results (percentages) was performed using a χ-square test on pooled data. All statistical analysis was performed using GraphPad Prism.

### Online supplemental material

Fig. S1 shows the effects of serum in culture medium on muscle fiber dedifferentiation, Fig. S2 shows contraction data in different concentration hydrogels, Fig. S3 shows example images used for myofiber classification, Fig. S4 shows sarcomere lengths in the different culture conditions, Fig. S5 shows contractile data after 10 days in culture, Fig. S6 shows the SDS-PAGE gel for identification of myosin heavy chain composition, and Fig. S7 shows contractile parameters in response to MT network manipulation.

## Results

### Isolated FDB muscle fibers atrophy, dedifferentiate, and lose contractile function during *ex vivo* culture in 2D

The culture of *ex vivo* muscle fibers is known to be limited because muscle fibers undergo structural and morphological

changes during culture (Smith and Meyer, 2020; Renzini et al., 2018). In this study, we sought to elucidate how muscle fiber morphology and function are affected in a 2D culture setting. To determine changes in muscle fibers during long-term culture, we isolated FDB muscle fibers and cultured them for up to 7 days (Fig. 1 A). First, we investigated the morphological changes by staining for α-actinin, myosin, and nuclei on days 1 and 7, which revealed sarcomeric aberrations and spreading of fiber ends following 7 days of *ex vivo* culture (Fig. 1 B), consistent with muscle fiber dedifferentiation (Brown and Schneider, 2002).

Muscle inactivity is known to cause muscle fiber atrophy and loss-of-function (Sayed et al., 2023). In our experimental setup, isolated muscle fibers are not stimulated during the culture period and thus stay inactive for the duration. To assess if skeletal muscle fibers may atrophy in culture, individual muscle fibers were tracked over 7 days, and their subsequent muscle fiber length and width were measured on days 1 and 7 after isolation. Although not significant, our measurements revealed a downward trend in muscle fiber length of ~5% following 7 days in culture (Fig. 1 C). Conversely, normalized muscle fiber width significantly decreased by ~15% following 7 days of culture (Fig. 1 D). To determine if long-term culture affected contractile function, we measured unloaded sarcomere shortening and contraction kinetics in response to electrical stimulation at days 1 and 7 after isolation. We observed a ~5% reduction in resting sarcomere length, going from 1.95 μm on day 1 to 1.85 μm on day 7 (Fig. 1 E). Similarly, sarcomere shortening dropped by ~50% over 7 days of the culture period (Fig. 1 F). Lastly, contractile velocity declined by ~60% over 7 days (Fig. 1 G). Taken together, these findings reveal that isolated muscle fibers atrophy and their morphology changes when cultured in 2D for 7 days, in which loss in muscle fiber width is more pronounced than its loss in length. In addition, these changes were accompanied by a significant decline in resting sarcomere length, sarcomere contraction, and contractile velocity.

### Low-serum conditions prevent dedifferentiation but do not rescue contractile function

To improve the usability of *ex vivo* cultured muscle fibers for long-term experiments, we aimed to prevent morphological alteration and preserve sarcomeric contraction. To do so, we first focused on the culture media composition, since high-serum media has previously been linked to muscle fiber dedifferentiation (Renzini et al., 2018; Selvin et al., 2015; Rosenblatt et al., 1995; Brown and Schneider, 2002). We compared low-serum culture medium with serum-free and high-serum (10% FBS) and evaluated their effects on muscle fiber morphology and contractile function. Microscopy imaging of high-serum cultures showed complete overgrowth of mononucleated secondary cell types, likely myogenic and fibrogenic cells (Fig. S1 A), and pronounced dedifferentiation (sarcomeric aberration and branching), especially at the muscle fiber tip as compared with the low-serum condition (Fig. 1 A) and serum-free conditions (Fig. S1 A). Low-serum and serum-free conditions showed less morphological alterations, in which slightly more spreading of the fiber ends was observed in low-serum culture. Next, contractile measurements revealed sarcomere length decreases over

time independently of serum in the culture medium, but higher serum content significantly reduced sarcomere length compared with low- and serum-free conditions (Fig. S1 B). Similarly, sarcomere shortening was also decreased over 7 days in all conditions, but in this case, there was no significant effect of high serum in the culture medium (Fig. S1 C). Contractile velocity values for these measurements showed greater variability, but a similar trend toward decrease was seen in all conditions, regardless of serum content (Fig. S1 D). Since resting sarcomere length was significantly decreased over time due to high-serum culture medium, but no significant effect was found in sarcomere shortening, we also investigated sarcomere length at peak contraction (Fig. S1 E). These results revealed that culture in high-serum medium significantly lowers the length to which sarcomeres contract. Because low-serum culture did not reduce sarcomere function compared with serum-free culture, we continued with low-serum medium for future experiments since it provides essential hormones, lipids, and trace elements that may be beneficial for muscle fiber function.

### Embedding muscle fibers in fibrin hydrogel increases viability, reduces dedifferentiation, and partially rescues contractile function

In their native environment, muscle fibers are densely packed and surrounded by an extracellular matrix (ECM) that functions as a supportive scaffold and provides mechanical cues (Purslow, 2002; Engler et al., 2006; Meyer and Lieber, 2011). During the isolation procedure, muscle fibers are removed from their ECM, causing isolated fibers to become unloaded. While these fibers will reattach to the culture surface, these attachments happen only in a single plane. The absence of ECM support and subsequent mechanical forces acting on the cell may induce morphological changes and atrophy of the fibers, leading to the deterioration of muscle fiber contraction. To determine the effect of the ECM on muscle fiber morphology and sarcomere contraction, we compared 2D cultured muscle fibers with those cultured in 3D hydrogels. Hydrogels have unique biomimetic properties and tunable mechanical properties dependent on the polymer type, polymer concentration, and cross-linking density that can help mimic the microenvironment of cells (Kopecek, 2007; Lin and Anseth, 2009). Thus, we developed a hydrogel matrix to embed fibers during culture and provide a 3D environment. This hydrogel was adapted from muscle tissue engineering protocols, which often use a mix of fibrin and basement membrane matrix solution (van der Wal et al., 2023; Rao et al., 2018). In this formulation, the fibrin supplies most of the structural support, while the added basement membrane matrix may help to facilitate connections between the fiber and the matrix (Fig. 2 A). First, we tested gels with increasing fibrin concentrations (2.5, 3.75, and 5 mg/ml fibrin) to determine if the cross-linking density would influence muscle fiber contractility, since a higher polymer concentration increases cross-linking density, resulting in a stiffer gel (Lin and Anseth, 2009). Fibrin concentrations did not negatively impact resting sarcomere length (Fig. S2 A) or contraction of embedded fibers (Fig. S2 B). Although not significant, a concentration of 5 mg/ml did seem to have a trend toward reducing contractile speed (Fig. S2 C). Based

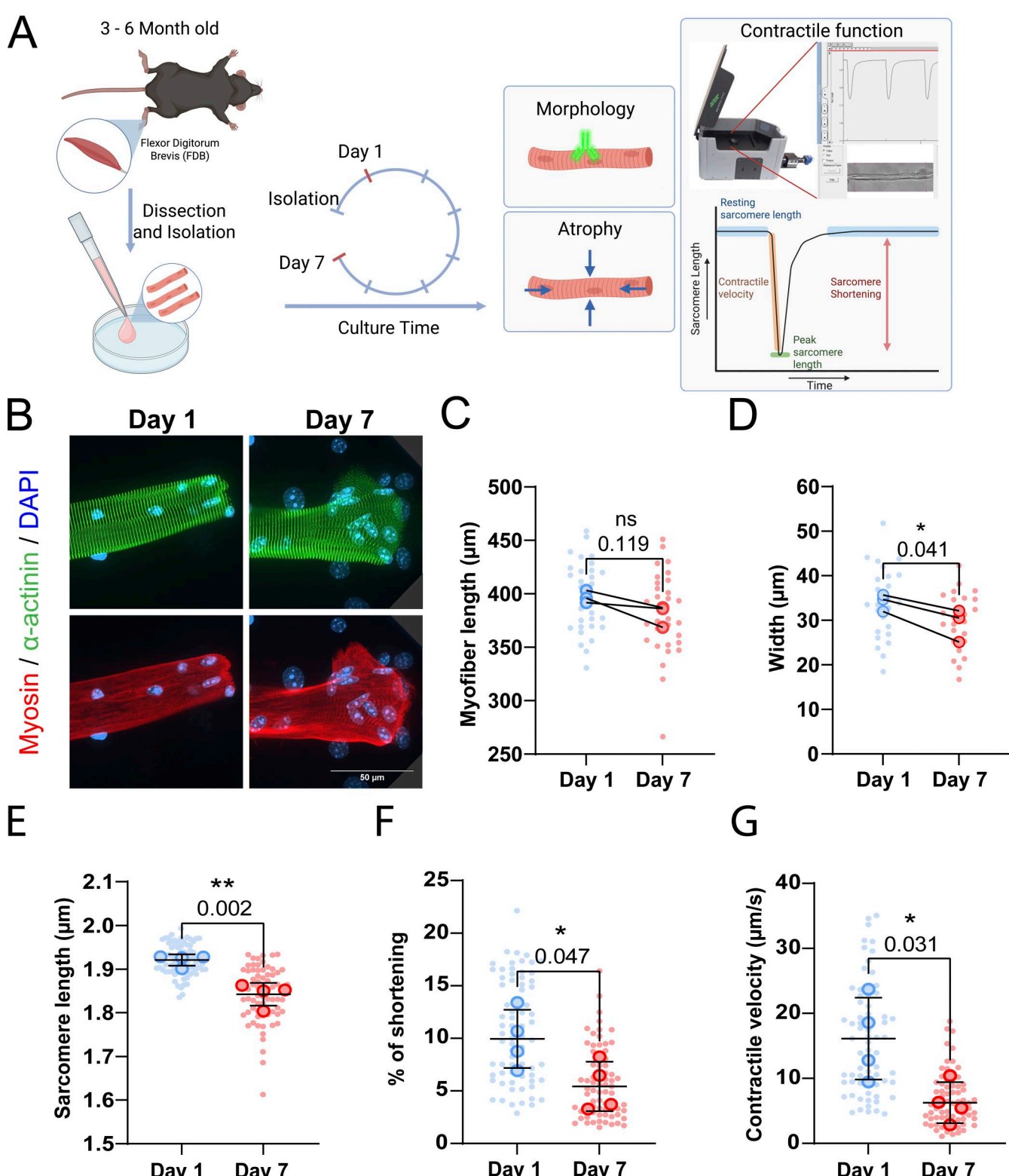

Figure 1. **Effects of culture time on muscle fiber morphology and function. (A)** Graphical overview of the culture experiment (image made in BioRender). **(B)** Representative confocal maximum intensity projection images taken with a 60× objective stained for α-actinin (green), myosin (red), and nuclei (blue) in muscle fibers cultured at days 1 and 7. Note the disordered sarcomere structure and spreading of the end of the fiber at day 7. **(C)** Muscle fiber length at days 1 and 7 of culture. **(D)** Muscle fiber width at day 1 and day 7 of culture. **(E–G)**: Contractile measurements of muscle fibers kept in culture for 1 and 7 days. **(E)** Quantification of muscle fiber resting sarcomere length at days 1 and 7 of *ex vivo* culture. **(F)** Quantification of the percentage of sarcomere shortening at days 1 and 7 of *ex vivo* culture. **(G)** Quantification of the contractile velocity of muscle fiber at days 1 and 7 of *ex vivo* culture. Data are means ± SEM; large dots represent mean values per mouse, and small dots represent single muscle fibers. Significance was determined using Student's *t* test with P < 0.05 considered significant with * = P < 0.05 and ** = P < 0.01.

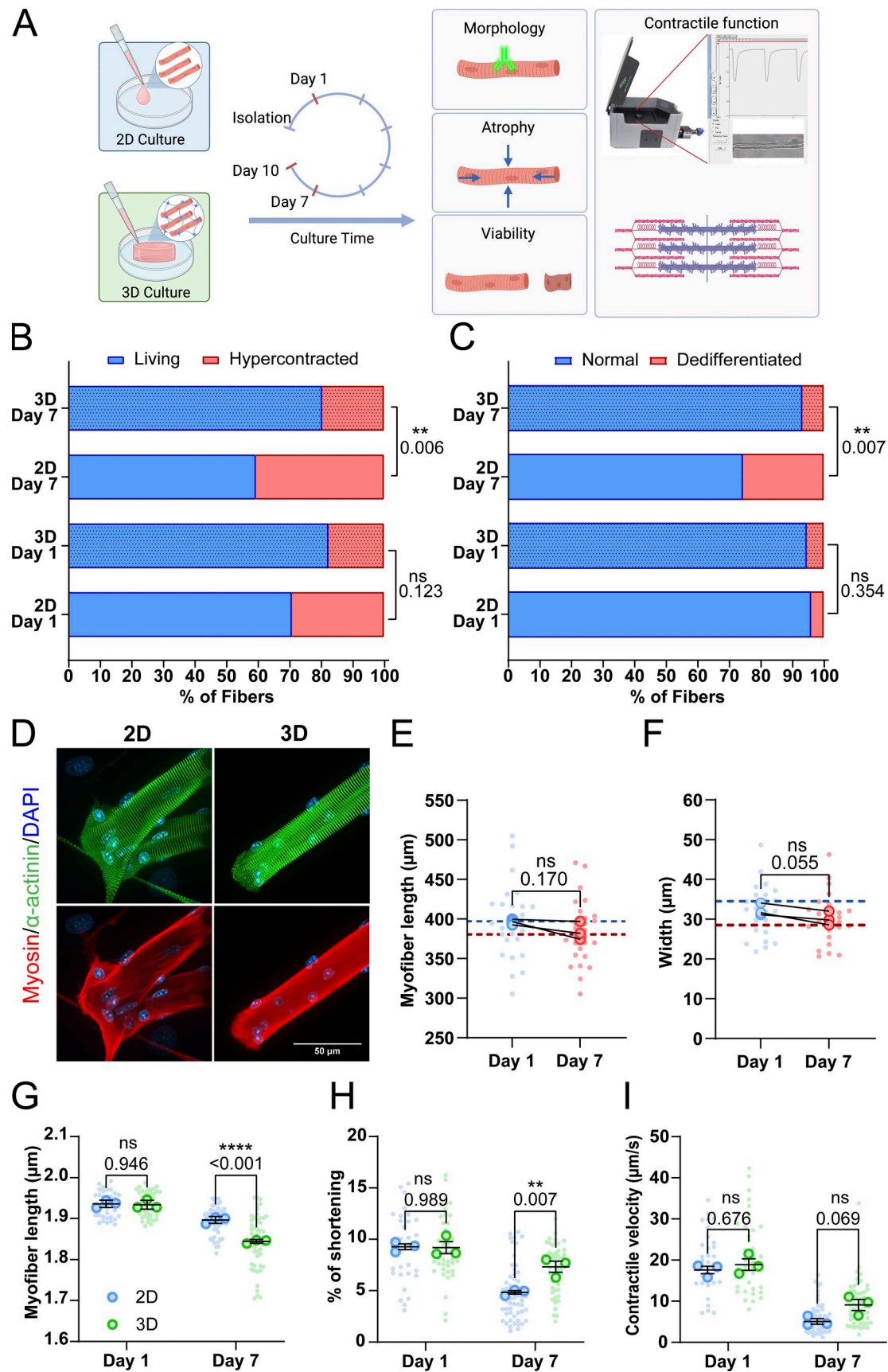

Figure 2. **Effects of embedding muscle fibers in a 3D fibrin/basement membrane hydrogel during culture. (A)** Graphical illustration of the 3D culture experiment (image made in BioRender). **(B)** Viability classification of muscle fibers cultured in 2D and 3D at days 1 and 7. **(C)** Dedifferentiation classification of

muscle fibers cultured in 2D and 3D at days 1 and 7. **(D)** Representative confocal maximum intensity projection images stained for α-actinin (green), myosin (red), and nuclei (blue) in muscle fibers cultured for 7 days in 2D and 3D. Note the lack of sprouting of the muscle fiber end in the 3D cultured muscle fibers. **(E)** Quantification of muscle fiber length cultured in 3D at days 1 and 7 of *ex vivo* culture. Dashed lines represent the mean length results of the 2D experiment in Fig. 1. **(F)** Quantification of muscle fiber width cultured in 3D at days 1 and 7 of *ex vivo* culture. Dashed lines represent the mean length results of the 2D experiment in Fig. 1. **(G–I)** Contractile measurements of muscle fibers kept in culture for 1 and 7 days in 2D and 3D culture systems. **(G)** Quantification of resting sarcomere length at days 1 and 7 cultured in 2D and 3D. **(H)** Quantification of percentage of sarcomere shortening at days 1 and 7 cultured in 2D and 3D. **(I)** Quantification of contractile velocity at days 1 and 7 cultured in 2D and 3D. Data are means ± SEM; large dots represent mean values per mouse, and small dots represent single muscle fibers. Significance was determined using χ-square test in panel B, Student's *t* test in panels E and F, and two-way ANOVA for panels G–I with $P < 0.05$ considered as significant with ** = $P < 0.01$ and **** = $P < 0.001$.

on these results, it was evident that any of the three fibrin gels could be used without affecting contraction parameters. To simplify the gel casting procedure, prevent premature solidification of hydrogels, and reduce starting material, we selected the 2.5 mg/ml fibrin concentration for the subsequent 3D experiments.

To explore if this 3D culture environment could improve the viability of isolated skeletal muscle fibers and reduce dedifferentiation, we assessed muscle fiber viability and dedifferentiation through blinded visual classification of cultured muscle fibers according to a representative image (Fig. S3, A and B). Results of fiber classification on day 1 show no significant change in the ratio of live/dead fibers between 2D and 3D (Fig. 2 B), as well as no difference in the amount of dedifferentiation (Fig. 2 C). Over time, we found that culture in 3D significantly increased the ratio of live/dead muscle fibers and reduced the amount of dedifferentiation, indicating a 3D environment promotes increased viability and stability of muscle fibers in culture (Fig. 2, B and C).

Next, we focused on the sarcomere structure of muscle fibers by staining for α-actinin, myosin heavy chain, and nuclei after 7 days of culture in either 2D or 3D conditions (Fig. 2 D). As shown previously, muscle fibers cultured in 3D lack most of the dedifferentiation (i.e., sprouting) and associated sarcomere disorganization at fiber ends that we observed in our 2D culture. Like in 2D (Fig. 1), we tracked single fibers cultured in 3D and measured their length (Fig. 2 E) and width (Fig. 2 F) at days 1 and 7. Here, we found no significant change in fiber length (Fig. 2 E). Similar to the 2D culture condition, fiber length exhibited a trend toward shorter muscle fiber length (Fig. 1 C). Although not significant, the normalized fiber width of fibers cultured in 3D showed a similar trend toward a decrease (Fig. 2 F) as the muscle fibers cultured in 2D (Fig. 1 D).

Lastly, we compared contractile parameters of muscle fibers kept in 2D and 3D culture conditions to assess how a 3D environment affects contractile function over time. On day 1, 3D culture had no significant acute effect on contractile function compared with 2D, indicating that any day 7 changes should result from the culture condition alone (Fig. 2, G–I). Over 7 days of culture, contractile function declined across all parameters in both 2D and 3D cultured fibers, with day 7 showing significant differences in resting length, sarcomere shortening, and contraction speed (Fig. 2, G–I). Resting sarcomere lengths were noticeably more reduced in muscle fibers cultured in 3D than those in 2D in MultiCell measurements (Fig. 2 G). Conversely, this decrease in sarcomere length was not observed in the experiments where the same fiber was tracked over 7 days, where

sarcomere length was significantly longer in fibers cultured in 3D than in those cultured in 2D (Fig. S4 A). Intriguingly, sarcomere shortening was reduced to a lesser extent in 3D compared with 2D, showing only a 25% decline as compared with a 50% decline in 2D (Fig. 2 H). Notably, sarcomere length at peak contraction was higher in 2D but decreased in 3D compared with day 1 (Fig. S4 B), while contractile speed measurements showed a minor, nonsignificant improvement in 3D (Fig. 2 I). Finally, to investigate whether our culture system could be used for studies that require even longer culture duration, such as viral modification combined with gene editing, we assessed contractile function after 10 days of culture in either 2D or 3D (Fig. S5). These results further illustrate the protective effect of 3D culture, as resting sarcomere length (Fig. S5 A) did not further decrease from day 7 in 3D cultures, while we observed a steep decline in the 2D culture. Sarcomere shortening (Fig. S5 B) and contractile velocity (Fig. S5 C) did not decrease further from day 7 in either 2D or 3D, with fibers cultured in 3D performing significantly better in both parameters. Taken together, these results illustrate that a 3D culture environment provides several benefits to muscle fibers. Namely, it improved viability, reduced dedifferentiation, and preserved contractile function.

## Cultured FDB muscle fibers lose contractile force due to atrophy in both 2D and 3D culture

Our experiments revealed that 3D culture conditions helped maintain the contractile performance of cultured muscle fibers. Since muscle fibers cultured in 2D undergo dedifferentiation, atrophy, and a reduction of sarcomere contraction, we hypothesized that the underlying sarcomeric structures become altered or damaged during 2D culture, ultimately impairing their contractile function. To investigate whether the changes in contractile function in cultured muscle fibers were due to damage or alterations to the sarcomere structure, we determined the maximal force-generating capacity in permeabilized FDB muscle fibers after culture. These measurements allowed us to directly assess sarcomere function, independent of other factors that may affect contraction, such as calcium handling, membrane excitability, and substrate adherence. To do this, muscle fibers were cultured for 1 and 7 days in 2D and 3D, after which fibers were permeabilized and attached to a length motor and force transducer. Fibers were stretched to a sarcomere length of 2.5 μm for optimal overlap between the thick and thin filaments, after which calcium-induced (pCa²⁺ of 4.5) sarcomere contraction was initiated (Fig. 3 A). Absolute maximal isometric force was then measured and normalized to fiber CSA. Absolute maximal force of muscle fibers significantly decreased after

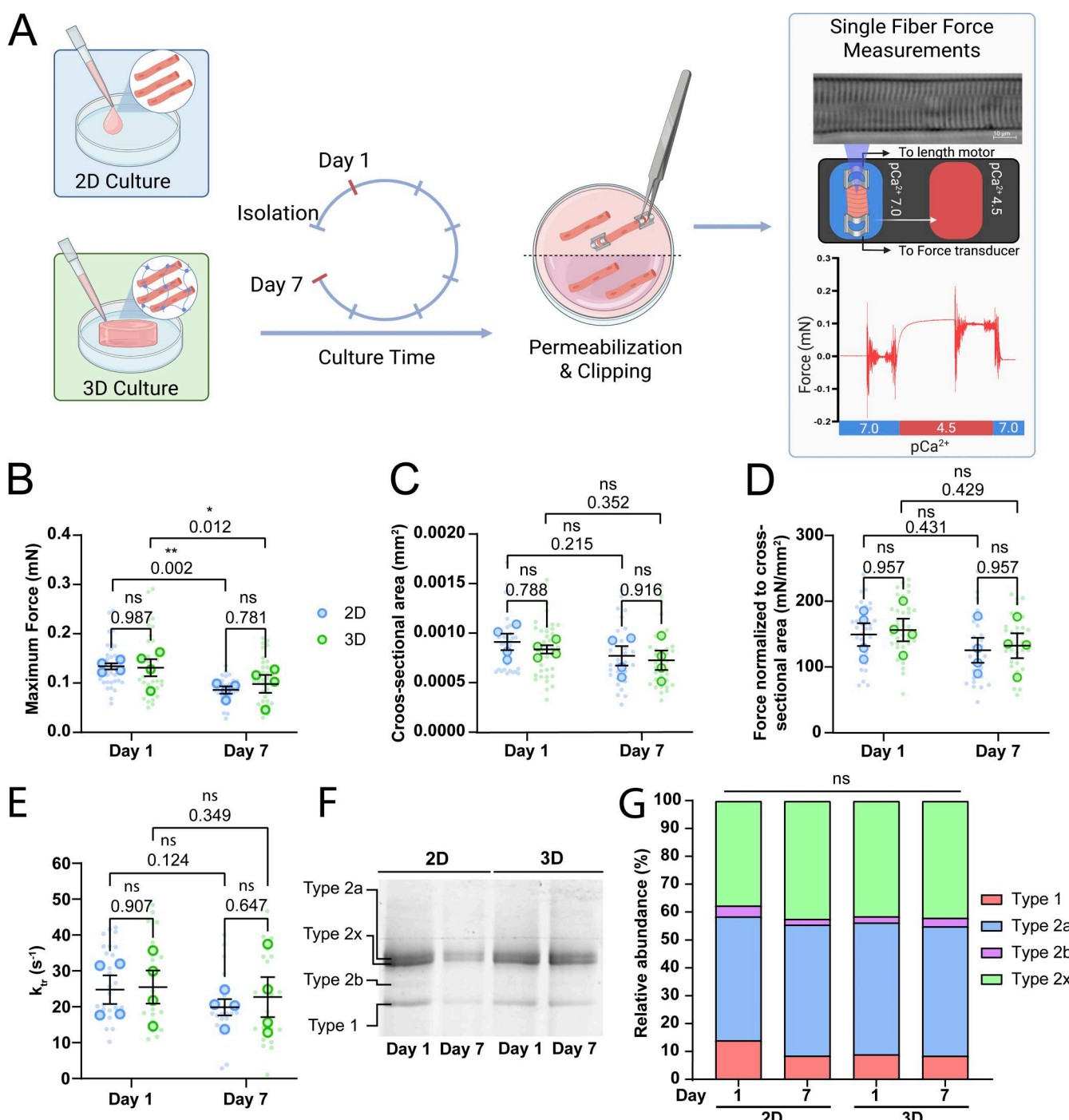

Figure 3. **Single muscle fiber force measurements of muscle fibers cultured in 2D and 3D systems on days 1 and 7 of culture. (A)** Graphical overview of the force measurement experiment (image made in BioRender). **(B)** Quantification of the maximum force production of skinned muscle fibers at days 1 and 7 cultured in 2D and 3D. **(C)** Quantification of muscle fiber CSA at days 1 and 7 cultured in 2D and 3D. **(D)** Quantification of the normalized maximum force to CSA at days 1 and 7 cultured in 2D and 3D. **(E)** Quantification of single fiber rate of tension redevelopment ($k_{tr}$) at days 1 and 7 cultured in 2D and 3D. **(F)** Example SDS-PAGE gel performed to analyze myosin heavy chain composition of pooled muscle fibers at days 1 and 7 cultured in 2D and 3D. **(G)** Quantification of relative myosin heavy chain abundance in pooled muscle fibers at days 1 and 7 cultured in 2D and 3D. Data are means ± SEM; large dots represent mean values per mouse, and small dots represent single muscle fibers. Significance was determined using the two-way ANOVA test in panels B–E and using χ-square test in panel G with P < 0.05 considered as significant with * = P < 0.05 and ** = P < 0.01. Source data are available for this figure: SourceData F3.

7 days of *ex vivo* culture, independent of culture condition, in which a comparison between 2D and 3D showed no significant effect on force generation of muscle fibers (Fig. 3 B). Next, muscle fiber CSA was measured. A slight downward trend in CSA was noticeable after 7 days of culture, independent of culture condition (Fig. 3 C), consistent with data from individual fibers tracked over culture time (Fig. 1 D and Fig. 2 F). After normalizing the force of each muscle fiber to its CSA, no

significant difference between days 1 and 7 was observed, which indicates that the decrease in absolute force may be driven by slight atrophy of the muscle fibers (Fig. 3 D). Additionally, we measured rates of tension development ($k_{tr}$) of cultured muscle fibers to further investigate crossbridge cycling. Rates of tension development were not significantly altered by culture in either 2D or 3D after 7 days (Fig. 3 E). Lastly, we pooled all measured muscle fibers and performed SDS-PAGE to investigate changes in myosin heavy chain composition (Fig. 3 F). These results show that there is no significant change in the distribution of the main myosin heavy chains within our samples (Fig. 3 G). Furthermore, protein bands for developmental myosin heavy chain isoforms, such as embryonic and neonatal, which should theoretically be found below type 1 bands and between type 2x and 2b bands, respectively, were not observed in our gel (Fig. S6). Our data suggest that long-term muscle fiber culture does not induce significant enough changes to the sarcomere to affect contractile force or tension development in both the 2D and 3D conditions. Thus, the preserved unloaded sarcomere contraction in 3D cannot fully be explained by direct effects on sarcomeric force, crossbridge cycling, or fiber type distribution.

### Reduction of contractile function in 2D is not caused by substrate adhesion

Since sarcomere organization, force production, and tension development in muscle fibers cultured in 2D and 3D were conserved (Fig. 3), the decrease in unloaded sarcomere shortening likely arises from a non-sarcomeric mechanism. Specifically, the discrepancy between preserved sarcomere organization (Fig. 3) and reduction in sarcomere shortening (Fig. 2) over time suggests that cells may become physically constrained over time. In addition, we observed that contractions of muscle fibers cultured in 2D were mainly impaired due to a reduction of sarcomere length at peak contraction (Fig. S3). Initially, we considered external constraints, since fibers cultured in 2D for 7 days showed sprouting of the fiber ends (Fig. 2 D), which may be a sign of increased adherence to the substrate. Adherence to the relatively stiff tissue culture plastic could constrain muscle fibers, leading to shorter contractions. Therefore, we hypothesized that adhesion to the substrate may negatively influence contraction.

To investigate the role of adhesion in contractile function, we first aimed to determine whether adhesion complexes could recover in culture, since collagenase-based digestion is known to destroy extracellular sarcolemma proteins (Gineste et al., 2022). To assess the presence and recovery of adhesion complexes, we stained muscle fibers for α-dystroglycan (Fig. 4 A) and quantified mean fluorescence intensity on days 1 and 7 in 2D and 3D (Fig. 4 C). Additionally, we also stained and quantified the intracellular membrane protein dystrophin to distinguish the effect of collagenase treatment on internal and external membrane proteins (Fig. 4, B and D). Our observations indicated that collagenase treatment disrupts the adhesion complex, which appears to be gradually reformed over the course of 7 days in both culture conditions. Although minimal protein recovery was noted on day 1, muscle fiber membranes were completely covered in α-dystroglycan on day 7. Between the fibers cultured in

2D and in 3D, no clear difference is visible in the costamere structure at day 7. Quantification of fluorescent signal also revealed that while α-dystroglycan is sparsely present in both groups at day 1, these complexes can completely recover by day 7 in both 2D and 3D culture conditions (Fig. 4 C). As expected, quantification of the dystrophin signal showed that internal membrane protein is not affected by this treatment (Fig. 4 D). Since the adhesion complex recovers equally in both 2D and 3D conditions, the presence or absence of adhesion complexes does not appear to explain the difference in shortening between 2D and 3D culture conditions.

To directly explore the role of adhesion in muscle fiber function, we cultured muscle fibers in adherent (lamin-coated) or suspension conditions for 7 days. Additionally, a second 2D condition was included, in which muscle fibers were detached from the culture dish using trypsin at day 6 and replated to mimic conditions at day 1 of culture (Fig. 4, D–G). Across all tested conditions, no significant differences were observed in contractile function. A slight downward trend is noticeable in the trypsin-treated fibers, likely due to the harsh effects of trypsin on muscle fiber integrity. Trypsinization can increase cell membrane permeability (Dias and Nylandsted, 2021), potentially causing calcium leakage in the cells, which may in turn affect the sarcomere length in muscle fibers. Our measurements reveal that enzymatically dissociating muscle fibers disrupts adhesion complexes, which are reformed over the course of 7 days. Additionally, whether fibers were adhered or not did not have any effect on the measured contractile parameters, suggesting that adhesion does not play a significant role in restricting contractile function in cultured muscle fibers, and this is consistent with our high-serum experiments that showed increased sprouting but no difference in sarcomere shortening (Fig. S1).

### Changes to MT organization impede contractile function of skeletal muscle fibers following prolonged culture

Our previous data demonstrate that changes in fiber health and function in culture cannot be easily explained by changes in sarcomere organization/function or external constraints due to cell-substrate adhesion. Thus, we explored internal constraints, since recent studies in cardiomyocytes have shown that non-sarcomeric cytoskeletal structures, such as MTs, can significantly influence contraction (Henderson et al., 2017; Robison et al., 2016; Caporizzo and Prosser, 2022). Building on previous results and the involvement of MTs in cardiomyocyte contraction, we sought to investigate if MTs play a role in the loss of contractile performance in cultured muscle fibers, taking advantage of the fact that the MT network can be readily manipulated with pharmacological agents, such as taxol (MT stabilization) and nocodazole (MT depolymerization). First, we treated muscle fibers with either nocodazole, taxol, or vehicle (control) at days 1 and 7. Fibers were fixed and MTs stained to determine the effectiveness of the treatment (Fig. 5 A). Control muscle fibers on day 1 showed an MT lattice structure typical of healthy muscle fibers, with longitudinal MTs running along the myofibrils and transversal MTs running along the z-discs (Belanto et al., 2016; Belanto et al., 2014). This lattice structure

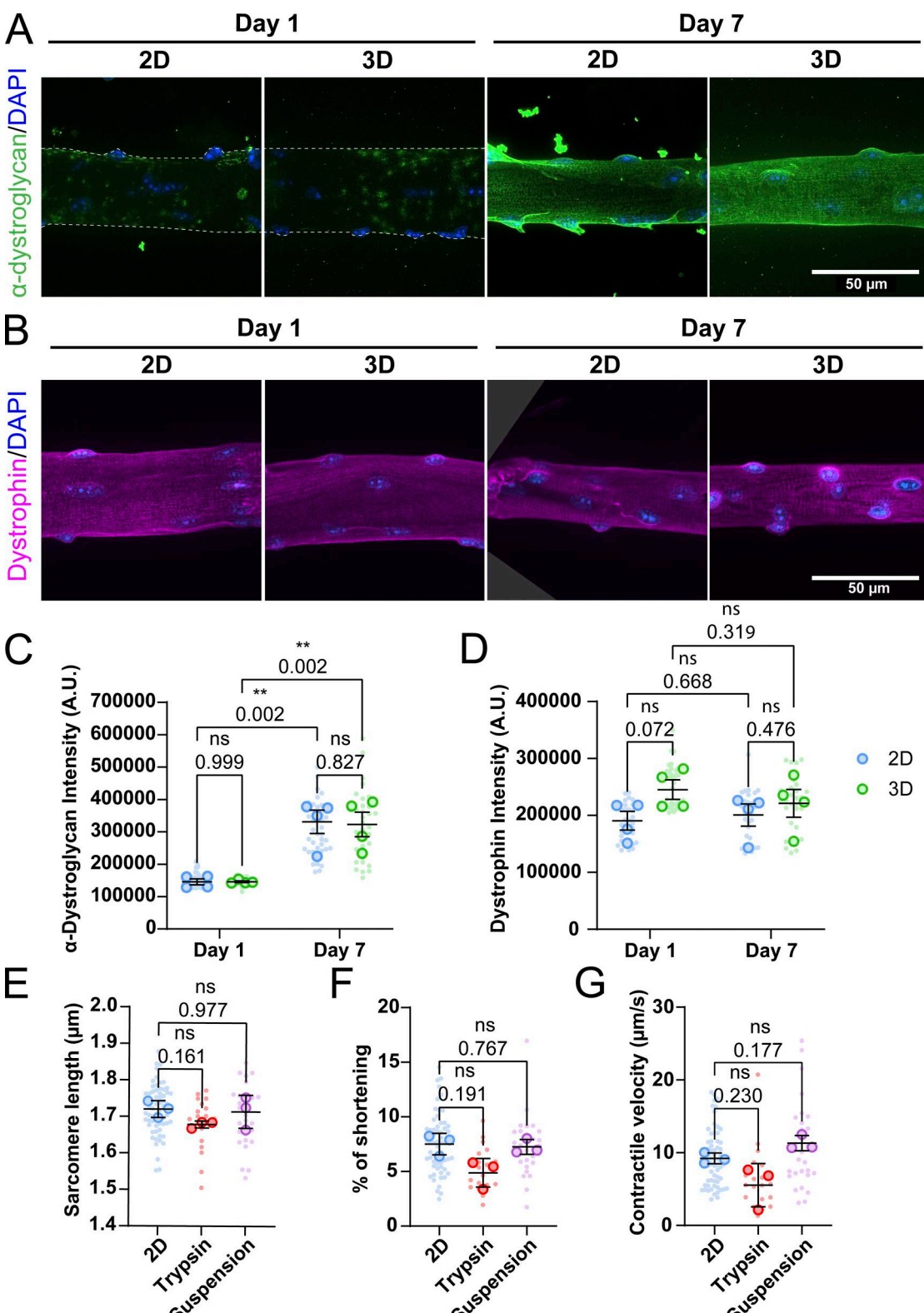

Figure 4. **Influence of muscle fiber adhesion on contractility. (A)** Representative confocal maximum intensity projection images taken with a 60× objective stained for α-dystroglycan (green) and nuclei (blue) in muscle fibers cultured for 1 and 7 days in 2D and 3D. Since treatment with collagenase destroys most membrane proteins, fibers are outlined with a dashed line at day 1. **(B)** Representative confocal maximum intensity projection images taken with a 60× objective stained for dystrophin (magenta) and nuclei (blue) in muscle fibers cultured for 1 and 7 days in 2D and 3D. **(C)** Quantification of mean α-dystroglycan fluorescence intensity in muscle fibers cultured for 1 and 7 days in 2D and 3D. **(D)** Quantification of mean dystrophin fluorescence intensity in muscle fibers cultured for 1 and 7 days in 2D and 3D. **(E–G)** Contractile measurements of muscle fibers cultured for 7 days in 2D, in 2D followed by dissociation with trypsin, or in suspension. **(E)** Quantification of resting sarcomere length of muscle fibers at day 7. **(F)** Quantification of the percentage of sarcomere shortening of muscle fibers at day 7. **(G)** Quantification of the contractile velocity of muscle fibers at day 7. Data are means ± SEM; large dots represent mean values per mouse, and small dots represent single muscle fibers. Significance was determined using one-way ANOVA with P < 0.05 considered as significant with ** = P < 0.01.

appears to be disorganized and denser after 7 days of culture (Fig. 5 A). As expected, muscle fibers treated with nocodazole showed almost complete depolymerization of the MT network, leaving only the nocodazole-resistant perinuclear stable MTs intact (Denes et al., 2021) (Fig. 5 A). Treatment with taxol had varying effects on fibers at days 1 and 7 of culture. On day 1, treatment seemed to cause disorganization of the MT network similar to control fibers at day 7. These changes were in line with our expectations since taxol is known to stabilize MT through posttranslational modification (PTM) and generate longitudinally bundled MTs (Mian et al., 2012). Nevertheless, treatment on day 7 seemed to partially restore a normal MT lattice structure, apart from the accumulation of MTs around the nucleus. The same interventions were performed in 3D cultured fibers, which revealed similar alterations in the MT network as those cultured in 2D (results not shown).

Next, we investigated if changes in the MT networks affected muscle fiber contraction at days 1 and 7 in both 2D and 3D conditions. At day 1 after isolation, neither nocodazole nor taxol had a significant effect on the sarcomere length, shortening velocity (Fig. 5, B–D), contraction time, relaxation time, and relaxation velocity (Fig. S7, A–C) values of muscle fibers in 2D or in 3D culture. On day 7, sarcomere length, contractile speed (Fig. 5, E and G), contraction time, relaxation time, and relaxation velocity (Fig. S7, D–G) were also not altered by nocodazole or taxol treatment in 2D and 3D culture. Interestingly, in the 2D condition, treatment with nocodazole on day 7 significantly increased muscle fiber shortening by 50%, from 5 to 7.5% (Fig. 5 F). Taxol treatment shows a similar trend, but this change was not significant. In the 3D culture condition, both nocodazole and taxol treatments at day 7 increased sarcomere shortening significantly, from ~7.5% to ~10%. These results indicate that changes in the MT network may contribute to a reduction in contractile function during prolonged muscle fiber culture, independent of whether cells are cultured in 2D or 3D. Strikingly, the combination of 3D culture and treatment with either nocodazole or taxol restored sarcomere shortening to a range found in muscle fibers on day 1 (Fig. 5 F). To summarize, we reveal that the MT network changes over time, independent of culture conditions. Furthermore, our contractile experiments revealed that changes to the MT network can significantly influence sarcomere contraction in cultured muscle fibers, providing evidence that the cytoskeletal network plays a prominent role in muscle fiber contraction.

## Discussion

In this study, we sought to gain a deeper understanding of culture-induced muscle fiber deterioration, as well as develop an improved culture system for *ex vivo* muscle fiber culture. To do this, we employed a recently developed optics-based, high-throughput contractile measurement system (Vonk et al., 2023). We first determined if contractile function decreases during culture at a similar rate as previously reported dedifferentiation of muscle fibers in culture (Renzini et al., 2018; Brown and Schneider, 2002; Rosenblatt et al., 1995; Keire et al., 2013; Duddy et al., 2011). In line with previous reports, muscle

fibers kept in culture for 7 days on laminin-coated dishes undergo dedifferentiation and show signs of atrophy in both muscle fiber length and width (Keire et al., 2013; Duddy et al., 2011). Dedifferentiation of muscle fibers *in vitro* may be attributed to an innate response to muscle injury (Mu et al., 2011). This process is exacerbated by the addition of serum to the culture medium and might be caused by growth factors present in the serum. Adding onto these findings, we reveal that the culture of muscle fibers also causes a reduction in sarcomere length and contractile function. In our long-term *ex vivo* culture setup, isolated muscle fibers remain unloaded and inactive for 7 days, resulting in a significantly lower sarcomere length than *in vivo* (Llewellyn et al., 2008). Chronic muscle shortening, in particular, can lead to the loss of serial sarcomeres, contributing to muscle atrophy (Adkins et al., 2021; Schiaffino, 2017). The likelihood that our cells are subject to chronic shortening stress, and consequently the loss of serial sarcomeres, is rather high. Despite this, our measurements of muscle fiber length and sarcomere length revealed a similar percentile decline in both, indicating that the reduction in muscle fiber length is likely driven by the reduction in sarcomere length, rather than by a loss of sarcomeres. Using our long-term *ex vivo* culture setup, we used contractile function to assess the quality of cultured muscle fibers. Although contractile function can be used to assess fiber quality, we found that loss of function is not intrinsically linked to dedifferentiation. In line with other studies, muscle fibers treated with high-serum medium underwent more dedifferentiation than those in low-serum medium (Renzini et al., 2018; Brown and Schneider, 2002; Rosenblatt et al., 1995). However, other than resulting in a reduced resting sarcomere length, the induced dedifferentiation did not exhibit a clear decrease in contractile function compared with fibers kept in low-serum conditions. These results suggest that the mechanism of dedifferentiation is not the main driver for functional decline in cultured muscle fibers.

Since loss-of-function is not directly linked to dedifferentiation, it is unlikely that alterations in the culture media alone can ameliorate these effects. In their native environment, muscle fibers are continuously exposed to contraction-induced forces and mechanical cues from their extracellular environment (Zhang et al., 2021; Csapo et al., 2020). However, during the isolation procedure, muscle fibers are denervated and removed from their environment, leaving them unloaded and not exposed to any mechanical forces. Additionally, mature muscle fibers do not spontaneously contract (Pham and Puckett, 2024), meaning they are not exposed to contractile forces that contribute to maintaining normal muscle function (van Ingen and Kirby, 2021). Because integrating contractile stimuli in our culture system proved difficult, we opted to focus on the effect of mechanical cues from the extracellular environment, which can be easily implemented in the *ex vivo* muscle culture platform as described in our previous work (Vonk et al., 2023). To test this, we embedded muscle fibers in a fibrin/basement membrane hydrogel to provide a scaffold with an anchoring surface. Our findings show that providing muscle fibers with a 3D environment increased the viability of muscle fibers while also preventing dedifferentiated morphology. In the study performed by

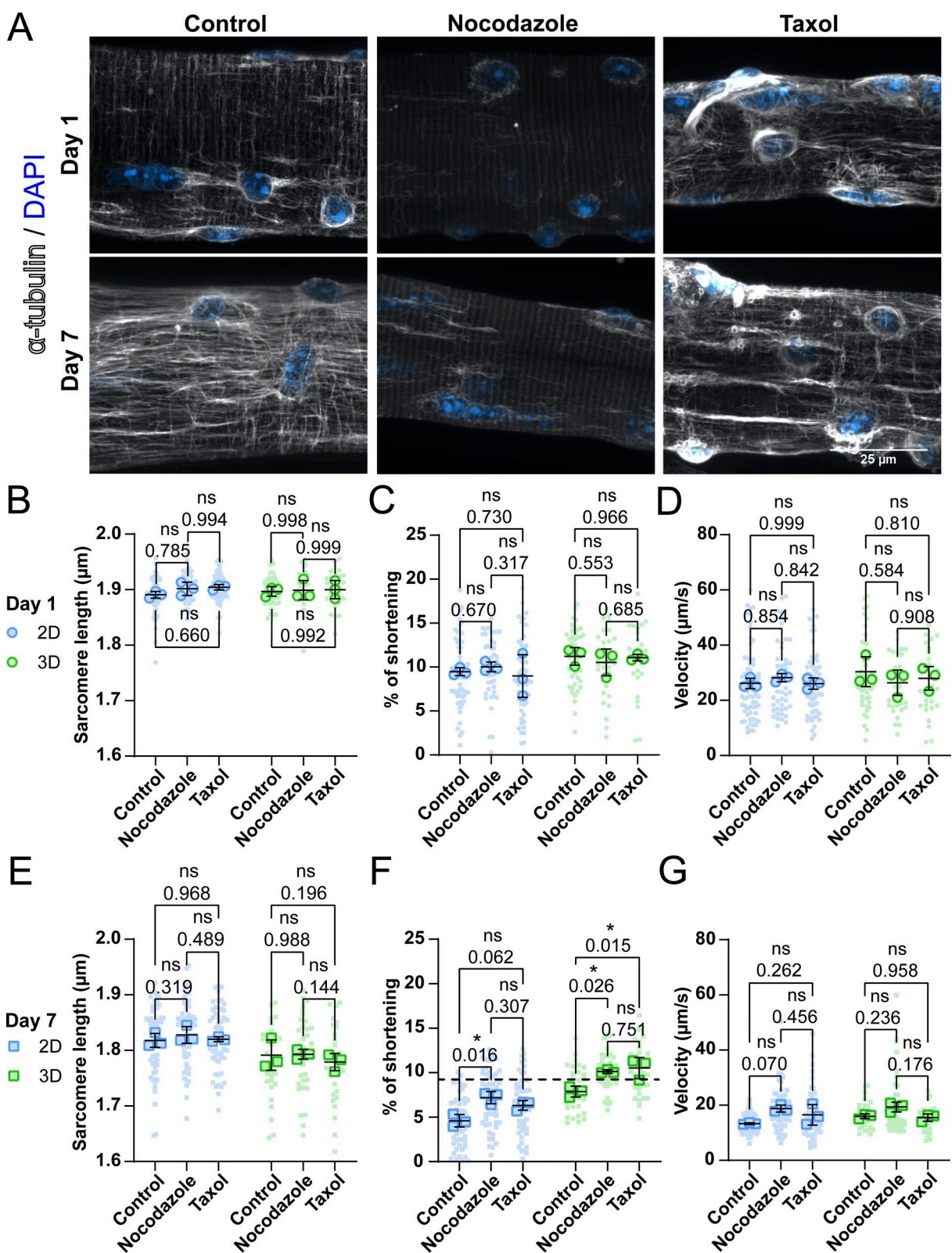

Figure 5. **MT networks in muscle fibers change during long-term *ex vivo* culture, thereby impairing sarcomere shortening. (A)** Representative confocal maximum intensity projection images taken with a 60× objective stained for showing α-tubulin (grey) and nuclei (blue) in muscle fibers cultured for 1 and 7 days in 2D and 3D. **(B–D)** Quantification of contractile measurements of muscle fibers cultured in 2D and 3D at day 1, treated with nocodazole, taxol, or untreated (control). **(E–G)** Quantification of contractile measurements of muscle fibers cultured in 2D and 3D at day 7, treated with nocodazole, taxol, or untreated (control). Data are means ± SEM; large dots represent mean values per mouse, and small dots represent single muscle fibers. Significance was determined using one-way ANOVA with P < 0.05 considered as significant with * = P < 0.05.

Brown and Schneider (2002), it was shown that embedding fibers in an agarose gel did not reduce the dedifferentiation of muscle fiber ends in culture. Since agarose is a bio-inert gel and does not provide an anchoring surface, it seems that both the 3D structure and attachment to the matrix play a role in reducing this dedifferentiation. Potentially, the sprouting observed in 2D culture conditions and in agarose gel may reflect a response to the loss of structural anchoring normally provided by tendon connections. Functional measurements revealed that providing this environment also helps to maintain contractile function over time. Initially, we theorized that the addition of a matrix might help retain sarcomere lengths in culture, since the matrix may provide an attachment surface that prevents muscle shortening. Resting sarcomere lengths of fibers cultured in 3D decreased significantly compared with those cultured in 2D when measured using the MultiCell system, but these findings were inverted when measuring sarcomere length in microscopy images. It is still not clear what caused this discrepancy, since sarcomere lengths were similar using both forms of measurements on day 1. A deeper look into sarcomere shortening revealed fibers cultured in 3D maintain a higher shortening since they exhibit significantly shorter sarcomere length at peak contraction when compared with the 2D condition. As of yet, it is still unclear what drives these changes and how this impacts fiber integrity.

To further explore the factors that drive this observed decline in muscle fiber contractility, we opted to determine if changes in function were caused by changes in sarcomeric force generation. By measuring force generation of permeabilized fibers cultured in 2D and 3D at different time points, we were able to assess if the changes were caused by changes at the sarcomere level. These measurements revealed that while absolute force generation of muscle fibers decreased over time, no changes between 2D and 3D cultured fibers were apparent. A potential reason for the observed decay in absolute force could be the consequence of inactivity of muscle fibers for 7 days. CSA measurements also revealed no significant atrophy in any of the measured groups, although a similar trend was seen as in previous measurements. This may have several causes, such as increased spread of data in this method or swelling of fibers due to permeabilization (Watanabe et al., 2019; Takemori et al., 2007). Nonetheless, when calculating the normalized force for each muscle fiber using these CSA measurements, no meaningful change in normalized force was observed in any of the groups. While a slight downward trend is seen, these results suggest that the contractile changes we see cannot be explained by a reduction of maximal force generation of the sarcomeres, particularly the differences observed between muscle fibers cultured in 2D and 3D.

Since the changes we observed could not fully be explained by force generation or increased substrate adhesion, we decided to investigate the role of the MT network in muscle fiber contraction. The involvement of MTs in sarcomere contraction has previously been demonstrated in cardiomyocytes, where the MT network was shown to play a crucial role in contractile dynamics by buckling and bearing load during contraction (Robison et al., 2016; Caporizzo and Prosser, 2022). Conversely, in skeletal

muscle fibers, global manipulations to the MT network appear to have limited effect on muscle fiber function (Kerr et al., 2015; Ai et al., 2003), which is consistent with our data at day 1 (Fig. 5, B–D). One unexpected finding of our study was that changes to the MT network at day 7 using either taxol and nocodazole were both able to improve sarcomere contraction. Immunofluorescent imaging revealed that the muscle fiber MT network is visibly altered during culture (Fig. 5 A). Since disruption of the MT network with nocodazole improved contraction only at day 7 and not day 1, we hypothesize that the observed alterations to the MT network on day 7 provide a specific constraint within the muscle fiber, thereby hindering contractile function. By removing this constraint with nocodazole, contractile function improves. More interestingly, we found that taxol was also able to increase contractions at day 7, despite having no significant effect on day 1. Thus, one hypothesis is that long-term culture of muscle fibers may lead to detrimental changes to the global organization or dynamics of the MT network, which can be alleviated with either nocodazole or taxol treatment, leading to similar improvements in contractile function. Whether the rescue effect of nocodazole and taxol is occurring through similar or distinct mechanisms, and whether the mechanism is direct, remains to be elucidated.

MTs can undergo PTMs, including acetylation and detyrosination, which have been shown to influence the contractile function of striated muscle cells (Kerr et al., 2015; Robison et al., 2016). In cardiomyocytes, an increase in MT detyrosination has been shown to increase coupling of MTs to sarcomeres, resulting in increased resistance and lowered contraction (Robison et al., 2016). Similarly, reducing MT detyrosination increases contractility in cardiomyocytes and, to a lesser extent, skeletal muscle fibers (Kerr et al., 2015). In addition, increasing acetylated MT decreases contraction in cardiomyocytes and skeletal muscle fibers (Coleman et al., 2021). These results suggest that the improvement of contractile function we observe on day 7 with taxol treatment is unlikely due to its effects of increasing detyrosinated or acetylated MT, which would be predicted to further impair contractility. Future studies should investigate the organizational changes that occur in the MT network during prolonged culture and whether these changes are driven by changes in the PTM profile to better understand the mechanisms by which MT dynamics and/or organization influence sarcomere contraction and overall muscle function in this system.

One limitation of our study is that we were unable to measure intracellular calcium concentrations during contraction. Several changes in cultured muscle fiber contractile mechanics can be explained by changes in calcium handling by muscle fibers. For example, changes in resting sarcomere length could be due to increased intracellular $Ca^{2+}$, while reductions in sarcomere shortening can be explained by reduced calcium release during contraction. Measurement of calcium transients proved to be technically challenging using the MultiCell system since it is optimized for cardiomyocytes instead of skeletal muscle fibers. Nevertheless, previous studies demonstrated that skeletal muscle fibers left in culture for up to 8 and 11 days, respectively, do not have altered $Ca^{2+}$ levels during both rest and contraction (Brown and Schneider, 2002; Ravenscroft et al., 2007). Since

these other studies report calcium handling does not change with long-term muscle fiber culture, it is unlikely that the decrease we see in contractile function after 7 days in culture is due to altered calcium transients. Nevertheless, we cannot rule out that changes in calcium handling are contributing to the increased contractile function in the 3D condition. Another limitation of this study is the complete inactivity and unloading of muscle fibers during culture. Inactivity, denervation, and unloading are all known causes for muscle fiber disuse atrophy, which could explain some of our findings (Nunes et al., 2022). Since disuse of muscle may lead to reduced contractile function, further improvements to the culture system could focus on combining the 3D culture system with electrical stimulation during prolonged culture. Using such a method, muscle fibers would receive additional mechanical signals, reducing the effects of disuse on fiber function. This approach would need to consider optimization of the stimulation parameters and media composition to limit pH changes that might occur from prolonged electrical stimulation. An alternative approach could be to use optogenetics to stimulate contraction, similar to recent reports in primary myoblast cultures (Hennig et al., 2023). Another possibility could be to introduce pneumatic stretch to the muscle fibers using flexible culture substrates. While such culture systems exist and show benefits in stem cell–derived cardiomyocytes (Kreutzer et al., 2020), passive stretch in muscle fibers should mostly be directed along the longitudinal direction of fibers, which is difficult to achieve in our culture system since muscle fibers are randomly aligned after the isolation procedure.

Long-term culture of explant muscle has long been a hurdle for *in vitro* skeletal muscle research. Since the first method of muscle fiber isolation was published in 1977 (Bekoff and Betz, 1977), several studies have looked into the effects of long-term culture of muscle fibers (Brown and Schneider, 2002; Ravenscroft et al., 2007; Pasut et al., 2013; Renzini et al., 2018; Rosenblatt et al., 1995). The results of these studies mainly illustrate the effects of serum in culture medium on muscle fiber morphology, revealing that proliferation medium with a high-serum content drastically increases dedifferentiation of muscle fibers (Renzini et al., 2018; Brown and Schneider, 2002; Rosenblatt et al., 1995). By combining *ex vivo* culture with high-throughput contractile measurements, we revealed how *ex vivo* culture affects muscle fiber contractility and how these changes are associated with muscle fiber dedifferentiation. Changes to the culture system, such as embedding muscle fibers in 3D, illustrated how improving the system can diminish culture-induced functional deterioration. Further development of this 3D culture system, for example tuning matrix stiffness to a level similar to *in vivo* muscle, may aid in maintaining muscle fiber contractile function *ex vivo*. Most strikingly, we revealed how MT structure changes over time and how these structural alterations play a key role in modifying muscle fiber contractile dynamics. Collectively, our data highlight the importance of providing muscle fibers with a 3D environment during *ex vivo* culture, particularly when testing pharmacological or genetic interventions to influence viability or contractile function.

## Data availability
The data underlying this study are available from the corresponding author upon reasonable request.

## Acknowledgments
Eduardo Ríos served as editor.

We would like to thank Sanna Luijcx, Nienke ten Cate, and Eva Larose for their help with muscle fiber isolation and culture. We would like to thank Sylvia Bogaards for the scientific discussion on *ex vivo* culture conditions, Emma Forde for performing the SDS-PAGE gel, and Stefan Conijn for analyzing the $k_{tr}$ data. Lastly, we would like to thank Michiel Helmes and Valentijn Jansen for their help and expertise in optimizing the MultiCell setup and analysis.

This work was funded by the National Institutes of Health (grant R01AR084850 to Tyler J. Kirby) and a research grant from A Foundation Building Strength for Nemaline Myopathies and from Prinses Beatrix Spierfonds (W.OR23-06) (to Coen A.C. Ottenheijm). Open Access funding provided by Vrije Universiteit Amsterdam.

Author contributions: Leander Vonk: conceptualization, data curation, formal analysis, investigation, methodology, project administration, resources, validation, visualization, and writing—original draft, review, and editing. Osman Esen: conceptualization, data curation, formal analysis, investigation, methodology, project administration, software, validation, visualization, and writing—original draft, review, and editing. Daan Hoomoedt: investigation. Rajvi Balesar: investigation. Coen A.C. Ottenheijm: conceptualization, project administration, supervision, and writing—review and editing. Tyler Kirby: conceptualization, funding acquisition, methodology, project administration, resources, supervision, and writing—review and editing.

Disclosures: The authors declare no competing interests exist.

Submitted: 13 January 2025

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

# Supplemental material

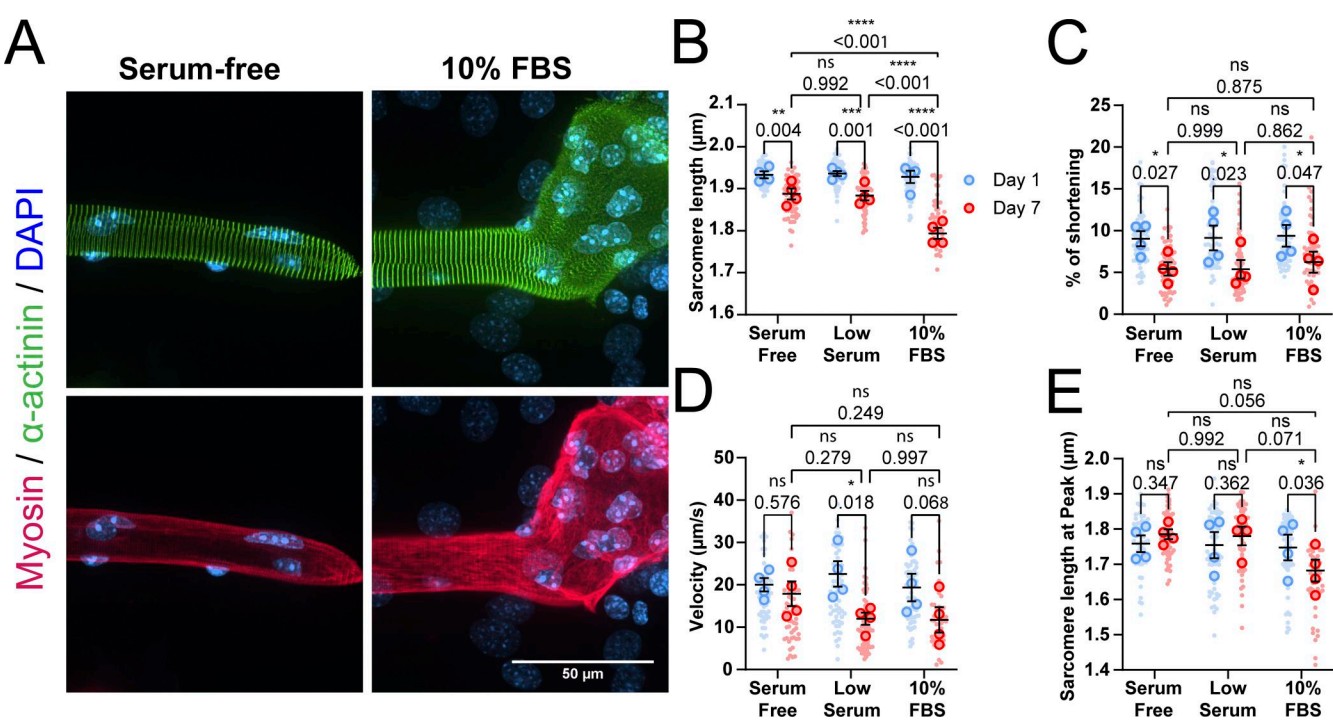

Figure S1. **Effects of serum in culture medium on muscle fiber dedifferentiation. (A)** Representative confocal maximum intensity projection images immunostained for α-actinin (green), myosin (red), and nuclei (blue) in muscle fibers cultured for 7 days in serum-free medium and medium containing 10% FBS. Note the lack of sarcomere misalignment in the muscle fiber cultured in serum-free medium and the large dedifferentiated mass in the muscle fiber cultured in 10% FBS. **(B–E)** Contractile measurements of muscle fibers cultured in low serum, medium containing 10% FBS, and serum-free medium, measured at days 1 and 7 of culture. **(B)** Quantification of resting sarcomere length at days 1 and 7 *ex vivo* culture. **(C)** Quantification of percentage of sarcomere shortening at days 1 and 7 *ex vivo* culture. **(D)** Quantification of contractile velocity at days 1 and 7 *ex vivo* culture. **(E)** Sarcomere length during maximal contraction of muscle fibers kept in culture. Data are means ± SEM; large dots represent mean values per mouse, and small dots represent single muscle fibers. Significance was determined using a two-way ANOVA with P < 0.05 considered as significant with * = P < 0.05, ** = P < 0.01, and *** = P < 0.001.

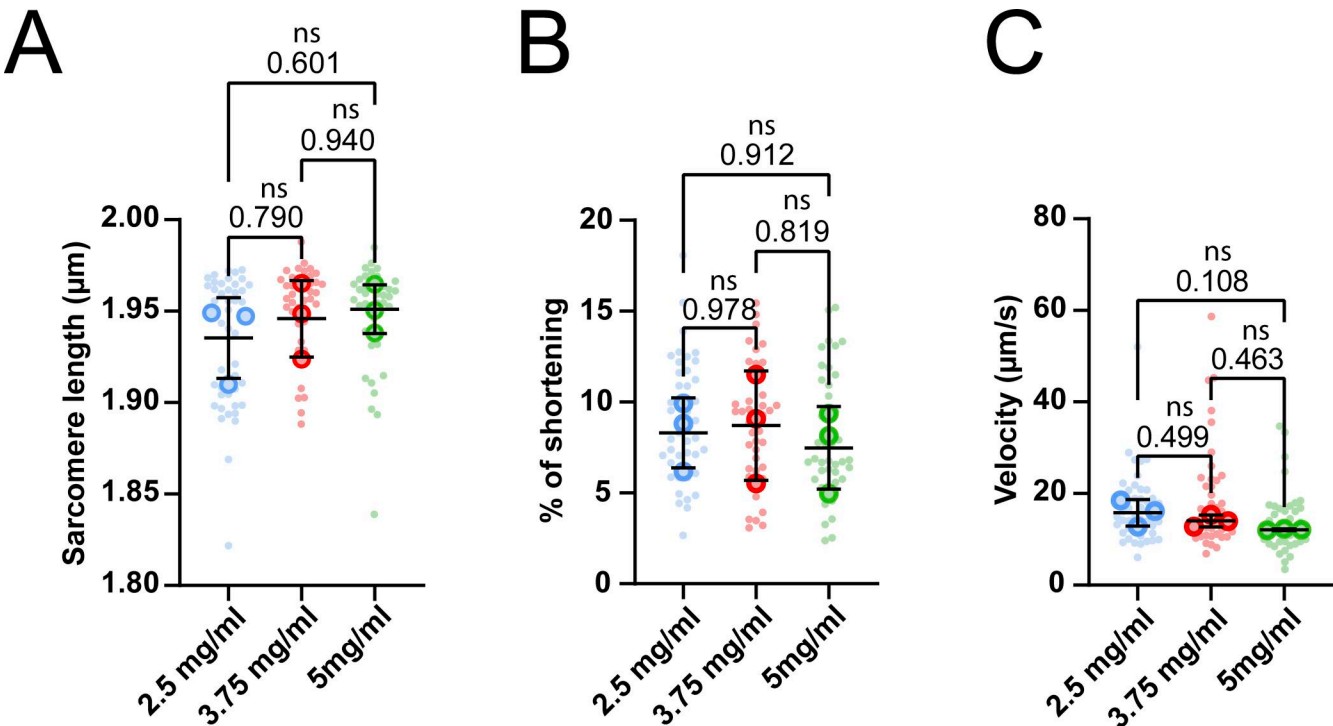

Figure S2. **Contractile measurements of muscle fibers in increasing 3D hydrogel concentrations, measured at day 1 of culture. (A)** Quantification of resting sarcomere length of muscle fibers in 3D hydrogel. **(B)** Quantification of sarcomere shortening of muscle fibers kept in 3D hydrogel. **(C)** Quantification of the contractile velocity of muscle fibers kept in 3D hydrogel. Data are means ± SEM; large dots represent mean values per mouse, and small dots represent single muscle fibers. Significance was determined using a one-way ANOVA with P < 0.05 considered significant.

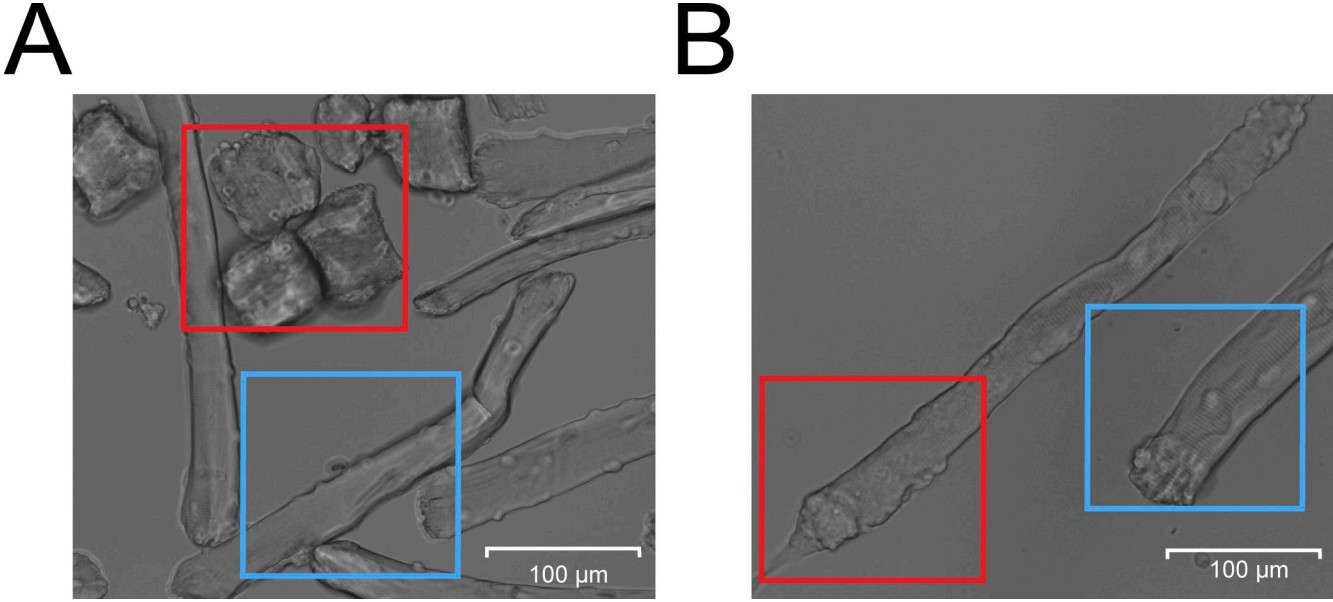

Figure S3. **Muscle fiber classification examples. (A)** Example image used for the classification of muscle fiber viability in which the red box includes dead/hypercontracted muscle fibers and the blue box includes living muscle fibers. **(B)** Example image used for the classification of muscle fiber dedifferentiation, in which the red box indicates a dedifferentiated muscle fiber and the blue box indicates a normal muscle fiber.

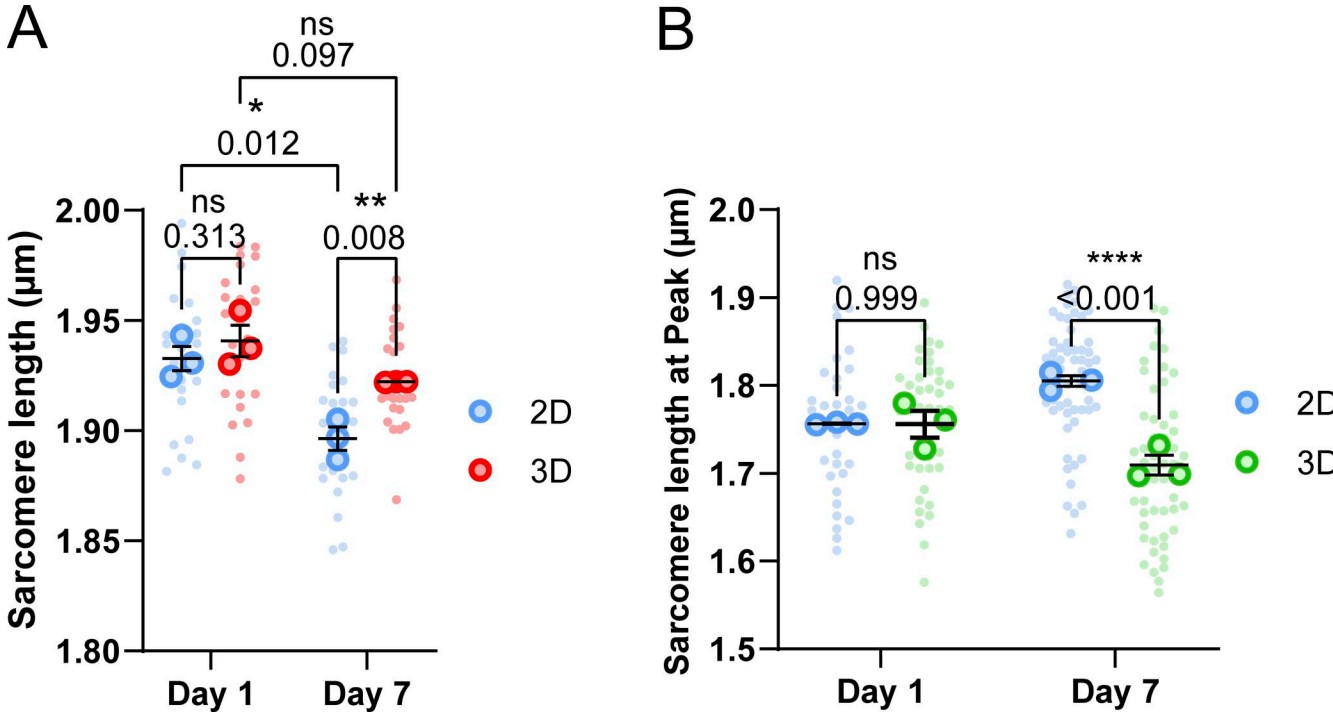

Figure S4. **Sarcomere length measurements of muscle fibers cultured in 2D and 3D culture systems at days 1 and 7 of culture. (A)** Quantification of resting sarcomere length of muscle fibers in 2D and 3D tracked over 7 days, which was used for normalization of muscle fiber width measurements in Fig. 1 D and Fig. 2 F. **(B)** Sarcomere length at peak contraction of muscle fibers cultured in 2D and 3D at days 1 and 7. Data are means ± SEM; large dots represent mean values per mouse, and small dots represent single muscle fibers. Statistics were performed using a repeated measure two-way ANOVA in panel A and two-way ANOVA in panel B, with $P < 0.05$ considered as significant with * = $P < 0.05$, ** = $P < 0.01$, and **** = $P < 0.001$.

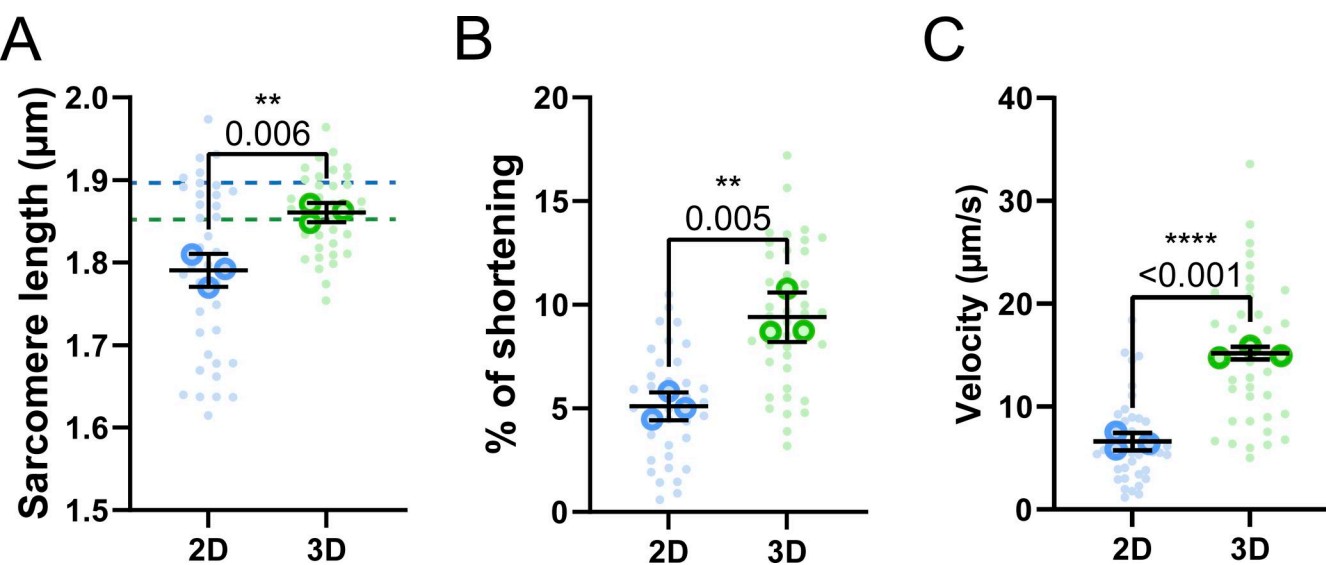

Figure S5. **Contractile measurements of muscle fibers cultured in 2D and 3D, measured at day 10 of culture. (A)** Quantification of resting sarcomere length of muscle fibers cultured for 10 days. The dashed line shows resting sarcomere length values of muscle fibers cultured for 7 days presented in Fig. 2. Note that the resting sarcomere continues to decrease in 2D, while it is maintained in 3D **(B)** Quantification of sarcomere shortening of muscle fibers cultured for 10 days. **(C)** Quantification of the contractile velocity of muscle fibers cultured for 10 days. Data are means ± SEM; large dots represent mean values per mouse, and small dots represent single muscle fibers. Significance was determined using Student's *t* test with $P < 0.05$ considered as significant with ** = $P < 0.01$ and **** = $P < 0.0001$.

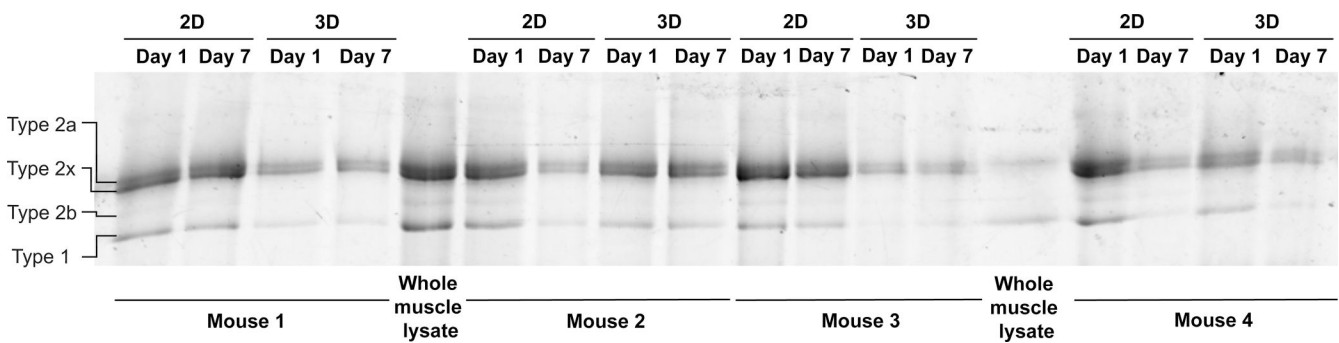

**Figure S6. SDS-PAGE gel for identification of myosin heavy chain composition.** SDS-PAGE separation of myosin heavy chain isoforms from pooled FDB muscle fibers of four mice cultured in 2D and 3D on days 1 and 7. Mixed whole muscle lysate from mouse soleus and EDL was used as a control in lanes 5 and 14. Source data are available for this figure: SourceData FS6.

**Figure S7. Contractile parameters in response to MT network manipulation in muscle fibers change during long-term *ex vivo* culture. (A–C)** Quantification of contraction, relaxation duration, and relaxation velocity measurements of muscle fibers cultured in 2D and 3D at day 1, treated with nocodazole or taxol, or untreated (control). **(D–F)** Quantification of contraction, relaxation duration, and relaxation velocity measurements of muscle fibers cultured in 2D and 3D at day 7, treated with nocodazole or taxol, or untreated (control). Data are means ± SEM; large dots represent mean values per mouse, and small dots represent single muscle fibers. Significance was determined using one-way ANOVA within the culture condition (2D or 3D), with P < 0.05 considered as significant.

