## [Peer Review File · The Journal of General Physiology]

Embedding muscle fibers in hydrogel improves viability and preserves contractile function during prolonged ex vivo culture

Leander Vonk, Osman Esen, Daan Hooimoedt, Rajvi Balesar, Coen Ottenheijm, and Tyler Kirby

Corresponding Author(s): Tyler Kirby, Amsterdam UMC Location VUmc

Review Timeline:

Submission Date:	January 13, 2025
Editorial Decision:	February 25, 2025
Revision Received:	September 16, 2025
Editorial Decision:	October 2, 2025
Revision Received:	October 8, 2025

Editor: Eduardo Ríos

Transaction Report:

DOI: <https://doi.org/10.1085/jgp.202513761>

February 25, 2025

Dr. Tyler Kirby
Amsterdam UMC Location VUmc
De Boelelaan 1108
Amsterdam 1081 HZ
Netherlands

Re: 202513761

Dear Dr. Kirby and co-authors.

Your manuscript has been evaluated by three expert referees and two editors, and discussed in detail by all JGP editors. I am happy to write that it was received positively and is potentially suitable for publication. There are, however, several concerns you must address before a decision can be reached, described in the referees' reports at the end of this letter. Please respond to them in detail.

Among them, I note especially those of referee #2, regarding your conclusion that excludes the sarcomeric structures as loci of the deleterious effects of culture, as well as the noted inconsistencies in the effects of the MT agonists. They question the assignment of any direct causation of the functional decay to changes in the MT network.

On the other hand, there is unanimous agreement about the potential value and applicability of the conditions that you devised for long-term culture (hydrogel, low-serum plus Taxol or Nocodazole).

Based on the contrasting evaluations of technical achievement and mechanistic insights, we suggest that you recast this manuscript as a Methods article, which would allow you to emphasize the technical aspects and qualify the mechanistic conclusions.

Additionally, the editors recommend that you "push" the duration of the culture beyond 7 days. Doing it would increase the value of the technique, as it would put the duration in a range useful for gene manipulation studies, among other applications. They also suggest running some protein gels to support the claims about dedifferentiation, i.e. reverting to expression of immature isoforms (a recommendation that would lose relevance if the study were communicated as a Methods article).

Whether or not the conversion of article category is done, the reviewers felt that the Discussion could be presented in a more compact manner, more closely linked to the actual results.

We hope that you will be able to submit a revised manuscript that addresses these points, which we believe will pose no problems, and which may be re-reviewed. In addition, please do not hesitate to contact me (via the editorial office) if you feel that a discussion of the reviewers' and editors' comments would be helpful.

Please submit your revised manuscript via the link below, along with a point-by-point letter that details your response to the reviewers' and editors' comments, as well as a copy of the text with alterations highlighted (boldfaced or underlined). If the article is eventually accepted, it would include a 'revised date' as well as submitted and accepted dates. If we do not receive the revised manuscript within one year, we will regard the article as having been withdrawn. We would be willing to receive a revision of the manuscript at a later time, but the manuscript will then be treated as a new submission, with a new manuscript number.

Please pay particular attention to recent changes to our instructions to authors in the following sections: Data presentation, Blinding and randomization and Statistical analysis, under Materials and Methods, as shown here: <https://rupress.org/jgp/pages/submission-guidelines#prepare>. Re-review will be contingent on inclusion of the required information (including for data added during revision) and demonstration of the experimental reproducibility of the results. Also, To improve the reproducibility of published content, we have partnered with SciScore. Authors are prompted in eJP to copy and paste the Materials and Methods section of their manuscript for a SciScore assessment when submitting their revised manuscript. Authors are encouraged (not required) to further revise their Materials and Methods if the SciScore is below 4. More information can be found here: <https://rupress.org/jgp/pages/submission-guidelines#sciscore>.

Please note, JGP now requires authors to submit Source Data used to generate figures containing gels and Western blots with all revised manuscripts (when applicable). This Source Data consists of fully uncropped and unprocessed images for each gel/blot displayed in the main and supplemental figures. If your paper includes cropped gel and/or blot images, please be sure to provide one Source Data file for each figure that contains gels and/or blots along with your revised manuscript files. File names for Source Data figures should be alphanumeric without any spaces or special characters (i.e., SourceDataF#, where F# refers to the associated main figure number or SourceDataFS# for those associated with Supplementary figures). The lanes of the gels/blots should be labeled as they are in the associated figure, the place where cropping was applied should be marked (with

a box), and molecular weight/size standards should be labeled wherever possible.

Source Data files will be made available to reviewers during evaluation of revised manuscripts and, if your paper is eventually published in JGP, the files will be directly linked to specific figures in the published article.

Source Data Figures should be provided as individual PDF files (one file per figure). Authors should endeavor to retain a minimum resolution of 300 dpi or pixels per inch. Please review our instructions for export from Photoshop, Illustrator, and PowerPoint here: <https://rupress.org/jgp/pages/submission-guidelines#revised>

Whilst you are revising your manuscript, we ask that you consider whether you have any artwork that might be suitable for the cover of JGP. Microscopy images are particularly good for cover artwork, but other types of image can be very effective, so we encourage you to be creative. Please don't restrict yourself to images from the paper; an image that is relevant to the work described would be just as suitable. Images should be a minimum resolution of 300 dpi. To see recent examples, visit the following page and click on 'Show covers? Yes': <https://jgp.rupress.org/content/by/year>

Thank you for submitting your interesting research to JGP.

Please submit your revised manuscript, and any associated files, via this link:

Link Not Available

Sincerely, Eduardo

Eduardo Ríos, Lic.

On behalf of Journal of General Physiology

Journal of General Physiology's mission is to publish mechanistic and quantitative molecular and cellular physiology of the highest quality; to provide a best-in-class author experience; and to nurture future generations of independent researchers.

Reviewer #1 (Comments to the Authors):

Vonk+Esen et al interrogate the mechanisms of skeletal muscle fiber dysfunction in ex vivo culture and the potential advantages of culturing in a 3D hydrogel. They find (both confirming and extending previous results) that in 2D culture adult muscle fibers atrophy, lost contractile function and dedifferentiate, and that contractile function and dedifferentiation can be partly or fully preserved by culturing in a 3D hydrogel. Further, they show that loss of contractile function in culture is not explained by reduced performance of myofilament or by altered adhesion in culture, but instead by culture-dependent alterations in the microtubule cytoskeleton. The paper has a clear rationale, is very well written and presented, and the primary conclusions are well supported by the data. The authors appropriately describe and caveat their results (and limitations), and the findings are of significance and use to muscle cell biologists and physiologists.

I have only minor recommendations and considerations to improve the depth of presentation and to more robustly support key conclusions and inform the field.

- 1) Most phenotypes are robustly quantified with appropriate methodology, except for immunofluorescence observations. Particularly for dedifferentiation, which the authors conclude is ameliorated (fully?) with 3D culture, but this is based on representative images. Perhaps a blinded scoring system (such as in 2C) could be utilized to quantify the % of fibers at different stages of differentiation (for example based on end branching/abnormalities).
- 2) Similarly, expression of α -dystroglycan levels from IF measurements should be quantified and Figure 4.
- 3) In lines 560-561 - I recommend referring to longitudinal MTs and transverse MTs (to avoid confusion with transverse (t)-tubules, as currently written)
- 4) The effects of taxol and nocodazole on Day 7 fibers are intriguing - but the effects of these two MT targeting drugs also appear different, which may point to distinct mechanisms of MT-dependent dysfunction and rescue - nocodazole appears to increase % shortening and velocity (which will be increased if shortening increases even if duration stays constant), but taxol increases shortening but with preserved velocity (which may actually indicate a longer duration contraction). Presumably with the multi-cell system the authors are able to extract contraction and relaxation durations (times)? If so they should report this data, as it may inform future interrogations into these mechanisms.
- 5) 3D culture promoted viability, reduced dedifferentiation, and helped preserve contractility. While not required to support the conclusions or establish the advance of this work, in future work it would be interesting to add electrical stimulation to the culture to see if this prevents the atrophy observed even in 3D culture.

Reviewer #2 (Comments to the Authors):

This MS explores possible mechanisms of muscle fibre deterioration in culture conditions. Muscle fibres were isolated by collagenase treatment and maintained in culture. Fibre survival was determined by the appearance of hypercontraction, by

measurements of fibre length and width, and by measurements of sarcomere shortening when stimulated. Structure was determined by microscopy coupled with staining of α -actinin and myosin. The performance of the contractile machinery was probed by permeabilising the fibres and measuring maximum Ca-activated force. The possible role of microtubules was investigated by staining with α -tubulin and using the depolymerizing agent nocodazole and the stabilizing agent taxol. The main variable explored were the concentration of serum in the culture medium and the difference between culture on laminin-coated plastic v. culture in an hydrogel/fibrin matrix. The study clearly shows that culture in hydrogel/fibrin improves structural survival and reduces the rate of deterioration of contractions and this appears to be a valuable contribution to the literature.

The study contains a wealth of detail and the main points are clearly made. Nevertheless I had difficulties in understanding the presentation at times and believe the MS could benefit from shortening and clarifying the presentation in places.

Abstract. I think the use of 2D and 3D as shorthand for culture on laminin-plated surface v. culture in a hydrogel/fibrin matrix is potentially misleading. I also think the statement that the loss of contractility can be partly explained by changes in the microtubular network overstates the evidence they provide.

Introduction. The first two paras are very general and could easily be deleted.

L221. Suggest adding a line explaining the purpose of adding aprotinin.

L257-259. Sentence doesn't seem to make sense. Maybe delete last 'with laminin.'

L294-296. If fibre volume is constant, and length decreases then fibre CSA will increase proportionately. One would predict that length decreases in proportion to CSA increase and in proportion to radius squared.

L297. How were the fibres removed from the hydrogel/fibrin matrix before preparation for permeabilization?

L364. I had considerable difficulty understanding this section for a variety of reasons. It might help if the running title was extended '.. dedifferentiation but the decline of contractility persists'. Part of the problem is the results are spread between Fig 1A and Fig S1A. I found Fig S1a confusing because in B,C,D & E the serum results are not in a sensible order. They should be Serum free, low serum, high serum. Furthermore the most important result, comparison of 1 to 7 days has no statistical significance mark. The most important result, that the decline in contractility persist in low serum is lost in L378 by an irrelevant linkage to the decline in resting sarcomere length. This section needs to be rewritten in straightforward language omitting the conclusions (L383-L385) which are better left to the Discussion.

L395 'leaving the isolated fibres completely unloaded'. Surely they are attached to the coverslip by laminin?

L481 While I agree that the observation that maximum force is preserved whilst stimulated contractions decline, is important I think your conclusion that the decline of contractility is not due to sarcomere function is premature. For example, shorting involves different crossbridge cycling to an isometric contraction and the proteins involved may have changed.

Fig 3A The image of a fibre at pCa 7 has some internal structure which I could not understand/resolve. If it is the sarcomere depicted why is it not in the the high Ca image? The image of the contraction is idealised. Could we have a real force record with solution changing artefacts etc. How is CSA area estimated?

L546 onwards. The section on microtubules is valuable but I think you overstate the mechanistic implications. My interpretation of your results is (i) that taxol and nocodazole had no effect on sarcomere shortening on day 1 despite having dramatic effects on MT structure and (ii) on Day 7 both nocodazole and taxol improved shortening. These facts seem to prevent any simple interpretation that MT are directly involved based on the data you provide.

Discussion. I enjoyed reading the discussion and felt it was a well balanced review of the situation. But it could easily be tighter and shorter and the teleological comments removed (L657-658).

Reviewer #3 (Comments to the Authors):

The authors of this manuscript provide a systematic investigation of the effect of contractile function during prolonged culture of embedding muscle fibers. Overall, the goal of this research was very well discussed and presented in this ms and the topic of this ms fit well with the scope of this journal. The quality of research (and their results) matched well for the scope of communication of this journal. The manuscript is well-structured and clearly written. Comparison of contractile function and morphological changes over the time between 2D and 3D culture were clearly demonstrated in this ms. Therefore, no further questions were raised with this manuscript.

We thank the editors and reviewers for their effort in reviewing our manuscript and for their insightful and thoughtful comments. We have addressed all of the reviewer comments in the revised manuscript, and we have included additional experimental data to further strengthen the conclusions of our manuscript. We believe that this has resulted in a significantly improved manuscript.

Comments from the editors:

Based on the contrasting evaluations of technical achievement and mechanistic insights, we suggest that you recast this manuscript as a Methods article, which would allow you to emphasize the technical aspects and qualify the mechanistic conclusions.

Additionally, the editors recommend that you "push" the duration of the culture beyond 7 days. Doing it would increase the value of the technique, as it would put the duration in a range useful for gene manipulation studies, among other applications. They also suggest running some protein gels to support the claims about dedifferentiation, i.e. reverting to expression of immature isoforms (a recommendation that would lose relevance if the study were communicated as a Methods article).

Whether or not the conversion of article category is done, the reviewers felt that the Discussion could be presented in a more compact manner, more closely linked to the actual results.

We thank the editors for their time and effort in reviewing our manuscript. We agree that extending the culture duration further would strengthen the value of this technique, so we performed a 10-day culture experiment for 2D and 3D culture of muscle fibers. Since these results revealed that 2D cultures are not viable for more than 10 days, we only included contractile measurements in (New Suppl. Figure 5). We found that at Day 10, resting sarcomere length continues to shorten in myofibers cultured in 2D, whereas this is not the case in 3D. Furthermore, the difference in contractile function that we observed at Day 7 is more pronounced at Day 10, indicating that the 3D culture system is beneficial for even longer studies.

Since reviewer #2 also had questions about myosin heavy chain isoform changes in our culture system, we performed an SDS-PAGE assay to identify myosin heavy chain isoforms in our cultured muscle fibers (New Fig. 3F,G and Suppl. Fig. 6). Here we found no change in myosin heavy chain isoform distribution, and the gel also did not indicate the expression of other isoforms such as neonatal or embryonic isoforms (which would theoretically be located between IIx and IIb, and below I, respectively). Since this process of muscle fiber dedifferentiation is still poorly understood, it is unclear whether dedifferentiation involves reverting to immature sarcomere isoforms or more general breakdown of the sarcomeres and cytoskeletal remodeling.

As per the suggestion of reviewer #2 and the editors, we have edited the discussion to be more concise. Lastly, we agree with the editors that this manuscript is best suited as a

methods article, which allows us to report our methods and findings in a more in-depth way, and we have resubmitted it as such.

Reviewer #1 (Comments to the Authors):

1) Most phenotypes are robustly quantified with appropriate methodology, except for immunofluorescence observations. Particularly for dedifferentiation, which the authors conclude is ameliorated (fully?) with 3D culture, but this is based on representative images. Perhaps a blinded scoring system (such as in 2C) could be utilized to quantify the % of fibers at different stages of differentiation (for example based on end branching/abnormalities).

We thank the reviewer for their comment and agree that quantification of differentiation would strengthen our findings. We have added another blinded scoring quantification for muscle fiber dedifferentiation (New Fig. 2C) and have moved the classification examples to Supplementary Figure 3A,B. This quantification shows that dedifferentiation events occur more often in 2D culture than in 3D.

2) Similarly, expression of α -dystroglycan levels from IF measurements should be quantified and Figure 4.

We originally did not include a quantification of α -dystroglycan since we found no significant effect of substrate adhesion in Figure 4, as well as no immediate visual differences. However, since our images showed that collagenase treatment seemed to completely disrupt α -dystroglycan, this may be the case for other sarcolemma proteins as well. In particular, this may affect the efficacy of the model in studies investigating receptors and other membrane proteins. As suggested by the reviewer, we have now quantified the α -dystroglycan fluorescence intensity levels (New. Fig. 4C). Alongside this, we have included staining and quantification for dystrophin to illustrate the difference between cytosolic and extracellular membrane-associated proteins (New. Fig. 4B,D), which showed that unlike α -dystroglycan, dystrophin expression is unaffected by the isolation procedure or culture time.

3) In lines 560-561 - I recommend referring to longitudinal MTs and transverse MTs (to avoid confusion with transverse (t)-tubules, as currently written)

We agree and this line has been changed.

“...with longitudinal MTs running along the myofibrils, and transversal MTs running along the Z-discs”

4) The effects of taxol and nocodazole on Day 7 fibers are intriguing - but the effects of these two MT targeting drugs also appear different, which may point to distinct mechanisms of MT-dependent dysfunction and rescue - nocodazole appears to increase % shortening and velocity (which will be increased if shortening increases even if duration stays constant), but taxol increases shortening but with preserved velocity (which may actually indicate a longer duration contraction). Presumably with the multi-cell system the authors are able to extract

contraction and relaxation durations (times)? If so they should report this data, as it may inform future interrogations into these mechanisms.

We agree with the reviewer that the results of the drug treatments are intriguing. We have now added the relaxation velocity and duration data to the manuscript (New Suppl. Fig. 7). The contraction duration data does show the expected outcome based on the change in % shortening and velocity - i.e. no change with nocodazole due to both % shortening and velocity both increasing, whereas taxol treatment showed a slight increase in duration due to increased % shortening with preserved contraction. Nevertheless, the duration data is more variable than the velocity data and further studies are needed to be able to draw meaningful conclusions or mechanistic implications from the contractile kinetics data.

5) 3D culture promoted viability, reduced dedifferentiation, and helped preserve contractility. While not required to support the conclusions or establish the advance of this work, in future work it would be interesting to add electrical stimulation to the culture to see if this prevents the atrophy observed even in 3D culture.

We agree with the reviewer that electrical stimulation during culture could potentially improve the culture system. This is something we have tried to implement in our system; however, this has proved to be technically challenging, as we suspect the repetitive electrical stimulation in our system results in cytotoxic gas and pH change in the medium, leading to an increase in cell death. To reflect this, the revised portion of the discussion related to this point now reads:

“Since disuse of muscle may lead to reduced contractile function, further improvements to the culture system could focus on combining the 3D culture system with electrical stimulation during prolonged culture. Using such a method, muscle fibers would receive additional mechanical signals, reducing the effects of disuse on fiber function. This approach would need to consider optimization of the stimulation parameters and media composition to limit pH changes that might occur from prolonged electrical stimulation. An alternative approach could be to use optogenetics to stimulate contraction, similar to recent reports in primary myoblast cultures (Hennig et al., 2023).”

Reviewer #2 (Comments to the Authors):

Abstract. I think the use of 2D and 3D as shorthand for culture on laminin-plated surface v. culture in a hydrogel/fibrin matrix is potentially misleading. I also think the statement that the loss of contractility can be partly explained by changes in the microtubular network overstates the evidence they provide.

We thank the reviewer for their comments and have changed the wording in the abstract to include more description of the 2D and 3D culture conditions. “on a laminin-coated culture dish (2D)” and “in a ... fibrin/Geltrex hydrogel (3D)”

We have also changed the statements about the MT network to reflect our findings better.
“We discovered that the loss of contractility of cultured muscle fibers was not the direct result of reduced sarcomere function but may be related to changes in the microtubule network.”

Introduction. The first two paragraphs are very general and could easily be deleted.

We agree with the reviewer and have removed the first paragraph for consistency and clarity. We have revised the second paragraph, as we find it important to give an overview of other *in vitro* models and their shortcomings compared to *ex vivo* culture of isolated muscle fibers.

L221. Suggest adding a line explaining the purpose of adding aprotinin.

We noted that aprotinin is added to prevent degradation of the fibrin gel. However, this may have been unclear, so we have added the purpose of using aprotinin.
“(a fibrinolysis inhibitor to prevent hydrogel degradation)”

L257-259. Sentence doesn't seem to make sense. Maybe delete last 'with laminin.'

The authors agree and this line has been changed.

L294-296. If fibre volume is constant, and length decreases then fibre CSA will increase proportionately. One would predict that length decreases in proportion to CSA increase and in proportion to radius squared.

We thank the reviewer for their comment and agree that if volume is constant and fiber length decreases, the CSA and thus diameter will have a proportional increase. We already included this in our analysis, as noted in the methods section, but it may have been worded confusingly. To clarify this point, we have now edited this section to read:

“To account for the fact that if fiber volume remains constant (i.e. isovolumetric), then sarcomere length and fiber width will change proportionally, we normalized width measurements to the percentage change in sarcomere length of the corresponding fiber. Thus, any observed change in fiber width would indicate a change in fiber volume.”

L297. How were the fibres removed from the hydrogel/fibrin matrix before preparation for permeabilization?

The muscle fibers were removed manually from either the gel or the 2D dishes using ultrafine forceps by simply peeling them out. This distinction has been added to the methods section.

“Single muscle fibers were then manually removed from the culture surface or hydrogel using ultrafine forceps,....”

L364. I had considerable difficulty understanding this section for a variety of reasons. It might help if the running title was extended '.. dedifferentiation but the decline of contractility

persists'. Part of the problem is the results are spread between Fig 1A and Fig S1A. I found Fig S1a confusing because in B,C,D & E the serum results are not in a sensible order. They should be Serum free, low serum, high serum. Furthermore the most important result, comparison of 1 to 7 days has no statistical significance mark. The most important result, that the decline in contractility persist in low serum is lost in L378 by an irrelevant linkage to the decline in resting sarcomere length. This section needs to be rewritten in straightforward language omitting the conclusions (L383-L385) which are better left to the Discussion.

We thank the reviewer for their comments and agree that the conclusions should be moved to the discussion section. We have added to the title to reflect our findings on contractility and have rewritten the section to be clearer and include the most important comparisons. We have changed the layout of Fig. S1 to a more logical order, as well as performing the statistics that were asked for in Fig S1.

The new title is: “Low-serum conditions prevent dedifferentiation but do not rescue contractile function”

New text: “Next, contractile measurements revealed sarcomere length decreases over time independently of serum in the culture medium, but higher serum content significantly reduced sarcomere length compared to low and serum-free conditions (Fig. S1B). Similarly, sarcomere shortening was also decreased over 7 days in all conditions, but in this case, there was no significant effect of high serum in the culture medium (Fig. S1C). Contractile velocity values for these measurements had a high amount of spread, but a similar trend toward decrease is seen in all conditions regardless of serum content (Fig. S1D). Since resting sarcomere length was significantly decreased over time due to high serum culture medium, but no significant effect was found in sarcomere shortening, we also investigated sarcomere length at peak contraction (Fig. S1E). These results revealed that culture in high serum medium significantly lowers the length to which sarcomeres contract.”

L395 'leaving the isolated fibres completely unloaded'. Surely they are attached to the coverslip by laminin?

Although fibers are attached to the coverslip after isolation, the digestion procedure causes all muscle fibers to become unloaded from their *in vivo* loaded state due to the loss of myotendinous attachment to the bone. Since the fibers are first unloaded, they will therefore only attach in their unloaded state and do not return to their *in vivo* loaded state. Additionally, any attachment on a coverslip will only feature attachments on a single side of the muscle fiber. The authors agree that this wording may be confusing and have changed the text accordingly.

“During the isolation procedure, muscle fibers are removed from their ECM, causing isolated fibers to become unloaded. While these fibers will reattach to the culture surface, these attachments happen only in a single plane.”

L481 While I agree that the observation that maximum force is preserved whilst stimulated

contractions decline, is important I think your conclusion that the decline of contractility is not due to sarcomere function is premature. For example, shorting involves different crossbridge cycling to an isometric contraction and the proteins involved may have changed.

We agree with the reviewer that this assertion may have been too premature. To investigate changes in cross-bridge cycling, we have also included rates of tension redevelopment (k_{tr}) measurements performed on these muscle fibers. Additionally, we have performed gel electrophoresis to determine changes in myosin heavy chain isoform composition of these samples. These results show that there is also no significant change in myosin heavy chain isoform composition or tension redevelopment. The new data has been incorporated into Figure 3 and the text has been toned down to remove premature conclusions.

Fig 3A The image of a fibre at pCa 7 has some internal structure which I could not understand/resolve. If it is the sarcomere depicted why is it not in the the high Ca image? The image of the contraction is idealised. Could we have a real force record with solution changing artefacts etc. How is CSA area estimated?

We thank the reviewer for their comment and have altered the representative image to show real force data and the corresponding fiber. The image of the muscle fiber included a stylized nucleus, but we have changed the figure to show only sarcomeres. The image shows a stylized view of the contractile measurement setup where a single muscle fiber is moved between baths filled with different calcium solutions. CSA of muscle fibers is estimated by measuring the width of the fiber in three locations from a standard top-down view, as well as the height of the fiber, which is viewed by a mirror to the side of the fiber. We have changed the methods section to clarify this.

L546 onwards. The section on microtubules is valuable but I think you overstate the mechanistic implications. My interpretation of your results is (i) that taxol and nocodazole had no effect on sarcomere shortening on day 1 despite having dramatic effects on MT structure and (ii) on Day 7 both nocodazole and taxol improved shortening. These facts seem to prevent any simple interpretation that MT are directly involved based on the data you provide.

We thank the reviewer for their comments. Previous reports indicate that acute microtubule depolymerization does not influence sarcomere shortening in isolated muscle fibers (Kerr et al., 2015) or force generation in whole muscle preparations (Ai et al., 2003). This is consistent with our Day 1 data, which, as the reviewer rightfully points out, microtubules seem to have minimal influence on contraction. Studies show that MT stabilization does not affect contractility and MT depolymerization does not alter peak force generation or calcium handling in healthy skeletal muscle fibers (Ai et al., 2003, Kerr et al., 2015). This suggests sarcomeres can generate force without immediate structural input from MTs. The improvement in contractile function at Day 7 using both nocodazole and taxol is intriguing and somewhat surprising, but we agree that the mechanism contributing to this rescue is still not known. We have rewritten the paragraph and toned down our conclusions to highlight that it is still not clear the extent to which microtubules are directly involved in the improvement of contraction at Day 7.

The new paragraph reads: “Since the changes we observed could not fully be explained by force generation or increased substrate adhesion, we decided to investigate the role of the MT network in muscle fiber contraction. The involvement of MTs in sarcomere contraction has previously been demonstrated in cardiomyocytes, where the MT network was shown to play a crucial role in contractile dynamics by buckling and bearing load during contraction (Robison et al., 2016, Caporizzo and Prosser, 2022). Conversely, in skeletal muscle fibers, global manipulations to the MT network appear to have limited effect on muscle fiber function (Kerr et al., 2015, Ai et al., 2003), which is consistent with our data at Day 1 (Fig. 5 B-D). One unexpected finding of our study was that changes to the MT network at Day 7 using either Taxol and Nocodazole were both able to improve sarcomere contraction. Immunofluorescent imaging revealed that the muscle fiber MT network is visibly altered during culture (Fig. 5A). Since disruption of the MT network with Nocodazole improved contraction only at Day 7 and not Day 1, we hypothesize that the observed alterations to the MT network on Day 7 provide a specific constraint within the muscle fiber, thereby hindering contractile function. By removing this constraint with Nocodazole, contractile function improves. More interestingly, we found that Taxol was also able to increase contractions at Day 7, despite having no significant effect on Day 1. Thus, one hypothesis is that long-term culture of muscle fibers may lead to detrimental organizational changes in the global organization or dynamics of the MT network, which can be alleviated with either Nocodazole or Taxol treatment, leading to similar improvements in contractile function. Whether the rescue effect of Nocodazole and Taxol is occurring through similar or distinct mechanisms, and whether the mechanism is direct, remains to be elucidated.”

Discussion. I enjoyed reading the discussion and felt it was a well balanced review of the situation. But it could easily be tighter and shorter and the teleological comments removed (L657-658).

We have reworded several sentences to remove teleological comments and shortened the discussion section slightly.

Reviewer #3 (Comments to the Authors):

The authors of this manuscript provide a systematic investigation of the effect of contractile function during prolonged culture of embedding muscle fibers. Overall, the goal of this research was very well discussed and presented in this ms and the topic of this ms fit well with the scope of this journal. The quality of research (and their results) matched well for the scope of communication of this journal. The manuscript is well-structured and clearly written. Comparison of contractile function and morphological changes over the time between 2D and 3D culture were clearly demonstrated in this ms. Therefore, no further questions were raised with this manuscript.

We thank the reviewer for taking the time to review our manuscript and for their positive evaluation.

Dr. Tyler Kirby
Amsterdam UMC Location VUmc
De Boelelaan 1108
Amsterdam 1081 HZ
Netherlands

Re: 202513761R1

Dear Dr. Kirby,

I am delighted to let you know that your manuscript, titled "Embedding muscle fibers in hydrogel improves viability and preserves contractile function during prolonged ex vivo culture" is scientifically acceptable for publication in Journal of General Physiology. Formal acceptance will follow when it is modified in accordance with the editor's remarks and our editorial policies.

In particular, clarify the meaning of the larger symbols in the graphs. We assume they represent mean values for each animal, but this should be pointed out in the legends.

Also, shouldn't your excellent diagram in Fig. 1 A and others reflect the extension of culture to 10 days?

Please note items that need attention are listed at the bottom of this email (under 'manuscript formatting checklist'). Explain whether and how you followed our last requests and highlight the modified text or figures if applicable. Your manuscript should be a double-spaced MS Word file and include editable tables, if appropriate.

Lastly, JGP requires a data availability statement for all research article submissions. These statements will be published in the article directly above the Acknowledgments. The statement should address all data underlying the research presented in the manuscript. Please visit the JGP instructions for authors for guidelines and examples of statements at <https://rupress.org/jgp/pages/editorial-policies#data-availability-statement>.

Please submit your final files via this link:

Link Not Available

Congratulations, to you and all co-authors, on this remarkable contribution. Thank you for choosing to publish your research in JGP and please feel free to contact me with any questions.

Sincerely,
Eduardo

Eduardo Ríos, Lic.
On behalf of Journal of General Physiology

Journal of General Physiology's mission is to publish mechanistic and quantitative molecular and cellular physiology of the highest quality; to provide a best in class author experience; and to nurture future generations of independent researchers.

Manuscript formatting checklist:

- MS Word document of text needed (including editable tables)
- MS Word document of supplemental text needed, if applicable (including figure legends and editable tables)
- Brief Statement describing supplementary information needed, if applicable (in subsection at end of Materials & Methods)
- Please include a data availability statement preceding the Acknowledgments section. Please see <https://rupress.org/jgp/pages/editorial-policies#data-availability-statement>
- Figures created at sufficient resolution and in acceptable format (including supplemental if applicable). If working in Illustrator, we prefer .ai or .eps file format. If working in Photoshop please use 600dpi/1000dpi .tiff or .psd file format. Minimum resolution at estimated print size: Minimum resolution for all figures is 600 dpi. For figures that contain both photographs and line art or text, 600 dpi is highly recommended. Figures containing only black and white elements (line art, no color, and no gray) should be 1,000 dpi. Maximum figure size is 7 in wide x 9 in high (17.5 x 22.8 cm) at the correct resolution. <https://jgp.rupress.org/fig-vid-guidelines>
- Supplemental figures, if any, conforming to same guidelines as manuscript figures (noted above)
- Source Data Figures should be provided as individual PDF files (one file per figure). Authors should endeavor to retain a minimum resolution of 300 dpi or pixels per inch. Please review our instructions for export from Photoshop, Illustrator, and PowerPoint here: <https://rupress.org/jgp/pages/submission-guidelines#revised>
- If images resemble one from a prior publications, the author must seek permissions (to reproduce or adapt) from the original

publisher. [You can resubmit your paper while waiting to hear back from the original publisher but please keep us updated]
- All authors must complete a disclosure form prior to acceptance. A link to complete the form has been sent to all coauthors.
Please provide the editorial office with updated email addresses if necessary

Reviewer #1 (Comments to the Authors):

The authors have appropriately responded to each of my considerations and have strengthened an already strong manuscript. I have no further concerns.

Reviewer #2 (Comments to the Authors):

The authors have responded appropriately to my comments in the first review.